# A microRNA program regulates the balance between cardiomyocyte hyperplasia and hypertrophy and stimulates cardiac regeneration

Andrea Raso[1,12], Ellen Dirkx[1,12], Vasco Sampaio-Pinto[1,2], Hamid el Azzouzi[1,3], Ryan J. Cubero [4,5], Daniel W. Sorensen[6], Lara Ottaviani[1], Servé Olieslagers[1], Manon M. Huibers [7], Roel de Weger[7], Sailay Siddiqi[8], Silvia Moimas [9], Consuelo Torrini[9], Lorena Zentillin[9], Luca Braga[9], Diana S. Nascimento [2], Paula A. da Costa Martins [1,10], Jop H. van Berlo [6], Serena Zacchigna [8], Mauro Giacca [9,11] & Leon J. De Windt [1✉]

Myocardial regeneration is restricted to early postnatal life, when mammalian cardiomyocytes still retain the ability to proliferate. The molecular cues that induce cell cycle arrest of neonatal cardiomyocytes towards terminally differentiated adult heart muscle cells remain obscure. Here we report that the *miR-106b~25* cluster is higher expressed in the early postnatal myocardium and decreases in expression towards adulthood, especially under conditions of overload, and orchestrates the transition of cardiomyocyte hyperplasia towards cell cycle arrest and hypertrophy by virtue of its targetome. In line, gene delivery of *miR-106b~25* to the mouse heart provokes cardiomyocyte proliferation by targeting a network of negative cell cycle regulators including E2f5, Cdkn1c, Ccne1 and Wee1. Conversely, gene-targeted *miR-106b~25* null mice display spontaneous hypertrophic remodeling and exaggerated remodeling to overload by derepression of the prohypertrophic transcription factors Hand2 and Mef2d. Taking advantage of the regulatory function of *miR-106b~*25 on cardiomyocyte hyperplasia and hypertrophy, viral gene delivery of *miR-106b~25* provokes nearly complete regeneration of the adult myocardium after ischemic injury. Our data demonstrate that exploitation of conserved molecular programs can enhance the regenerative capacity of the injured heart.

[1] Department of Molecular Genetics, Faculty of Science and Engineering, Faculty of Health, Medicine and Life Sciences, Maastricht University, Maastricht, The Netherlands. [2] i3S - Instituto de Investigação e Inovação em Saúde, INEB - Instituto Nacional de Engenharia Biomédica, ICBAS - Instituto de Ciências Biomédicas de Abel Salazar, University of Porto, Porto, Portugal. [3] Department of Molecular Genetics, Erasmus University MC, Rotterdam, The Netherlands. [4] The Abdus Salam International Centre for Theoretical Physics, Trieste, Italy. [5] IST Austria, Klosterneuburg, Austria. [6] Stem Cell Institute and Lillehei Heart Institute, Department of Medicine, University of Minnesota, Minneapolis, MN, USA. [7] Department of Pathology, University Medical Center Utrecht, Utrecht, The Netherlands. [8] Department of Cardiothoracic Surgery, Radboud University Medical Center, Nijmegen, The Netherlands. [9] International Centre for Genetic Engineering and Biotechnology (ICGEB), Trieste, Italy. [10] Department of Physiology and Cardiothoracic Surgery, Faculty of Medicine, University of Porto, Porto, Portugal. [11] School of Cardiovascular Medicine and Sciences, King's College London, London, UK. [12] These authors contributed equally: Andrea Raso, Ellen Dirkx. ✉email: l.dewindt@maastrichtuniversity.nl

Proliferation of cardiogenic precursor cells and division of immature cardiomyocytes is key to mammalian cardiac morphogenesis during embryonic and fetal development[1,2]. The neonatal mammalian heart still retains considerable proliferative capacity reminiscent of lower vertebrates[3–7], and is sustained by hypoxic conditions[8], activation of Hippo signaling[9], various transcriptional regulators[10–12], and endogenous microRNA mechanisms[13,14]. Complete regeneration of the injured myocardium, including the human neonatal heart[15], exclusively takes place in a short time window during fetal and early postnatal life, when cardiomyocytes still possess the ability to proliferate[7]. Shortly after birth, however, the majority of cardiomyocytes lose their proliferative capacity, and shift toward a terminally differentiated phenotype, with restricted ability to reactivate mitosis[16]. Limited myocyte turnover does occur in the adult mammalian heart, but insufficient to restore contractile function following injury[17]. The default growth response of the adult heart to overload or injury consists primarily of cardiomyocyte hypertrophy, a form of cellular growth without cell division that often precipitates in heart failure[18], a serious clinical disorder that represents a primary cause of morbidity and hospitalization. In humans, despite remarkable progresses made by device-based therapies and drug interventions[19], the only way to replace lost cardiomyocytes is heart transplantation. Identification of the developmental molecular mechanisms that stimulate a proliferative phenotype in early postnatal life, control cell cycle arrest, and produce a hypertrophic response of the terminally differentiated heart muscle, may hold the key to unlock the regenerative potential of the adult mammalian heart.

## Results

### Suppression of the *miR-106b~25* cluster facilitates cardiac remodeling.
We recently demonstrated that mature *miR-25* transcripts are downregulated in two mouse models of heart failure, and results in the derepression of the bHLH transcription factor *Hand2* in the postnatal mammalian myocardium[20]. *miR-25* is embedded in the *miR-106b~25* cluster, located on mouse chromosome 5 (chromosome 7 in humans) in an intronic region of the DNA-replication gene *Mcm7* and consists of three miRNAs: miR-106b, miR-93, and miR-25 (Fig. 1a). Here we show that each *miR-106b~25* cluster member displayed decreased expression in human cardiac biopsies of end-stage heart failure obtained upon heart transplantation compared to healthy controls (Fig. 1b), decreased expression in the calcineurin transgenic mouse model of heart failure (Supplementary Fig. 1a) and in pressure overloaded mouse hearts in both the early and late remodeling phase (Supplementary Fig. 1b, c).

To more directly address the ramifications of reduced expression of *miR-106b~25* for the postnatal heart, we subjected cohorts of *miR-106b~25*−/− mice[21], where expression of the cluster members was undetectable in the heart (Supplementary Fig. 1d), to sham surgery or pressure overload by transverse aortic constriction (TAC) surgery and serially assessed cardiac geometry and function at four weeks (Fig. 1c). Remarkably, *miR-106b~25*−/− mice already suffered from a mild form of eccentric hypertrophy at baseline as evidenced in sham-operated mice by an increased cardiac geometry, reduced wall thickness, reduced ejection fraction (EF), and increased cardiomyocyte size at four weeks after sham surgery (Fig. 1d–k; Table 1). Four weeks of TAC surgery in wild-type mice or *miR-106b~25*−/− mice resulted in severe myocyte disarray, interstitial fibrosis, increased heart weight, left ventricular dilation, systolic and diastolic dysfunction as well as a transcript induction of *Nppa*, *Nppb*, *Acta1*, and *Myh7* as hypertrophic "stress" markers with consistently more severe phenotypes in *miR-106b~25*−/− mice (Fig. 1d–k; Table 1). In line, silencing of the individual *miR-106b~25* cluster members with specific antagomirs (Supplementary Fig. 2a, b) resulted in spontaneous and mild eccentric remodeling as evidenced by an increased cardiac geometry (Supplementary Fig. 2c), abnormal echocardiographic parameters (Supplementary Fig. 2d–g), increased cardiomyocyte size (Supplementary Fig. 2h) and induction of hypertrophic markers (Supplementary Fig. 2j), albeit milder compared to the phenotype of *miR-106b~25*−/− mice, supporting the conclusion that the individual cluster members work in a concerted fashion.

Mechanistically, we have previously shown *miR-25*-dependent regulation of the prohypertrophic embryonic transcription factor Hand2 in the adult heart and could repeat this finding (Supplementary Fig. 2k, l)[20]. To further understand the mechanistic role of *miR-106b~25* in cardiac remodeling, we analyzed bioinformatics databases to search for *miR-106b*, *miR-93*, and *miR-25* binding sites in cardiac expressed protein-coding transcripts. We identified a perfect match for a *miR-25* heptametrical seed sequence in the transcription factor Myocyte enhancer factor 2d (*Mef2d*) that showed complete evolutionary conservation among vertebrates (Fig. 1l), validated the target gene using luciferase reporters harboring either the wild-type or site-directed mutagenesis of key nucleotides in the *miR-25* binding site (Fig. 1m), and finally demonstrated derepression of Mef2d protein expression upon *miR-25* silencing in cardiomyocytes by western blotting (Fig. 1n). Taken together, these data demonstrate that the *miR-106b~25* cluster is repressed in the failing heart and causes eccentric hypertrophic remodeling by simultaneous derepression of the prohypertrophic transcription factors Hand2 and Mef2d.

### Activation of the *miR-106b~25* cluster provokes cardiac enlargement.
To further evaluate the function of the *miR-106b~25* cluster in cardiac disease, we measured the expression of *miR-106b*, *miR-93*, and *miR-25* in the postnatal mouse heart at postnatal day 0 (p0), p5, p10, p15, and p20, representing developmental stages toward adulthood (p56, week 8). The data demonstrate that the *miR-106b~25* cluster was on average 6–8 fold higher expressed early in the postnatal phase and slowly decreased in expression to adult expression levels after which expression remained stable (Fig. 2a). In line, *miR-106b~25* expression in the adult heart muscle cells was very low compared to adult non-myocyte cells that still retain the ability to proliferate (Supplementary Fig. 3).

Next, we made use of the high cardiac tropism and prolonged expression of serotype 9 adeno-associated viral (AAV9) vectors following systemic delivery[22]. AAV9 vectors expressing *mmu-miR-106b*, *mmu-miR-93*, and *mmu-miR-25* precursor miRNAs (AAV9-miR-106b~25), or a control vector with an empty multiple cloning site (AAV9-MCS), were injected intraperitoneally in neonatal mice at postnatal day 1 (p1; Fig. 2b) as a gain-of-function approach (Fig. 2c). Anticipating a cardiac phenotype resistant to hypertrophic remodeling, much to our surprise, at 4 weeks, the hearts of mice injected with AAV9-miR-106b~25 were histologically normal but significantly enlarged (Fig. 2d, e). Echocardiographic analysis demonstrated that heart size and interventricular septum thickness in systole was significantly increased (Fig. 2e–i, Table 2), but cardiac morphometric dimensions and contractile function were normal as evidenced by the absence of ventricular dilation (Fig. 2h) and ejection fraction (Fig. 2i). There was no sign of inflammatory cell infiltration or cardiac fibrosis (Fig. 2d, j). Even more peculiar, individual myocyte size was unaltered and there was no evidence for reactivation of a "fetal" gene expression pattern characteristic for hypertrophic cardiomyocytes (Fig. 2k, l). Conclusively, maintaining high *miR-106b~25* expression levels as observed in

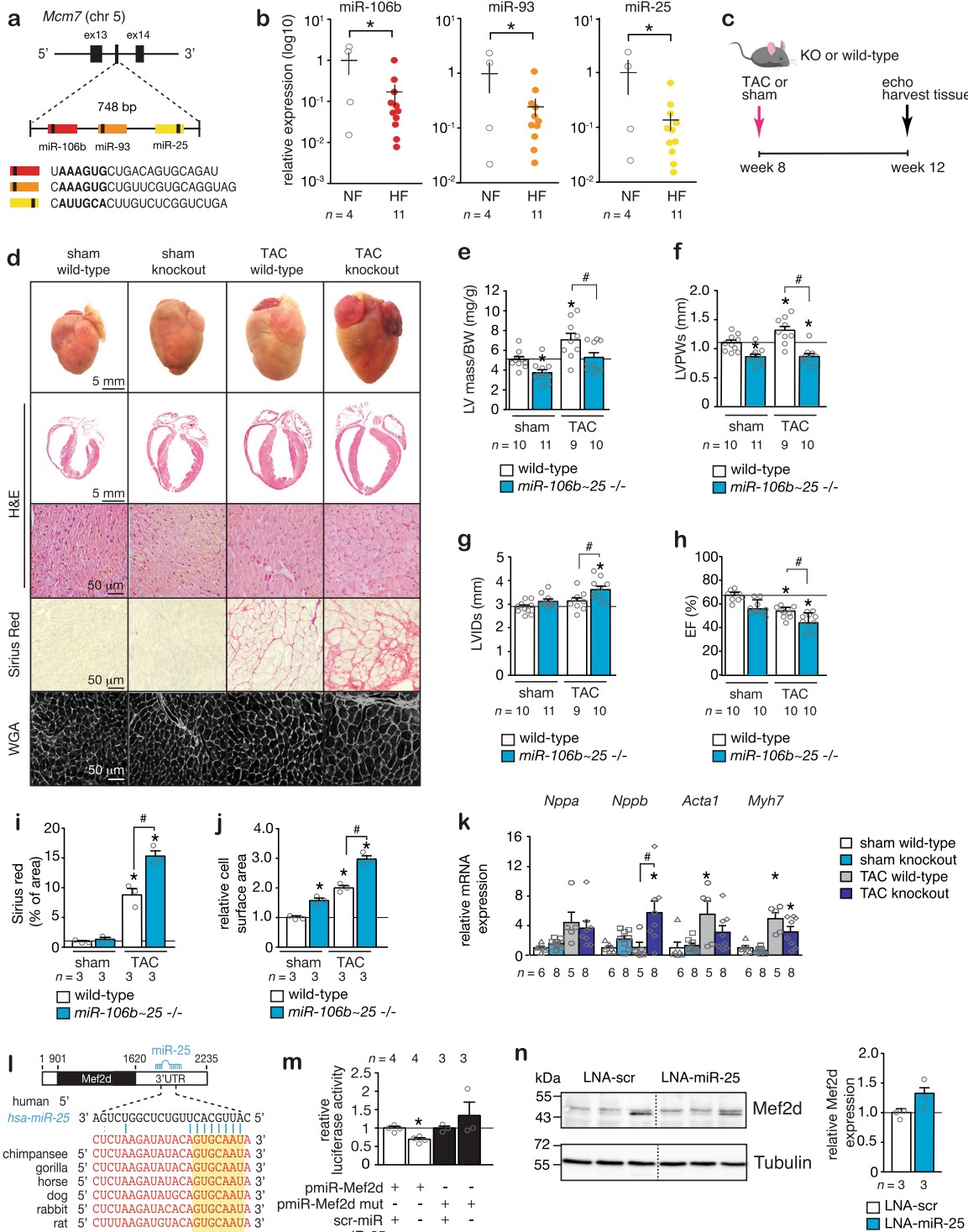

**Fig. 1 *miR-106b~25* gene deletion induces hypertrophic cardiac remodeling. a** A schematic representation of the mouse *Mcm7* gene harboring the *miR-106b~25* cluster in intron 13. **b** Real-time PCR analysis of *miR-106b*, *miR-93*, and *miR-25* abundance in human non-failing or failing myocardium. **c** Design of the study. **d** Representative images of whole hearts (top panels), haematoxylin & eosin (H&E)-stained sections of four-chamber view (second panel), high magnification H&E sections (third panel), Sirius Red stained sections (fourth panel), and wheat germ agglutinin (WGA)-stained (fifth panel) histological sections. Quantification of **e** left ventricular mass/body weight (BW) ratio, **f** left ventricular posterior wall thickness in systole (LVPWs), **g** Left ventricular internal diameter in systole (LVIDs), **h** ejection fraction (EF). **i** Quantification of the fibrotic area by Sirius Red staining and **j** cell surface areas by wheat germ agglutinin (WGA) staining. **k** Real-time PCR analysis of *Nppa*, *Nppb*, *Acta1*, and *Myh7*, *n* refers to number of animals. **l** Location and evolutionary conservation of *hsa-miR-25* seed region on Mef2d. **m** Activity assay of luciferase reporter constructs shows the binding of *hsa-miR-25* to the 3′UTR of Mef2d, *n* refers to number of transfection experiments. **n** Western blot analysis of endogenous Mef2d and GAPDH as a loading control in hearts from wild type (WT) versus *miR-106b~25* knock-out (KO) mice, *n* refers to number of hearts. *$P < 0.05$ vs corresponding control group; #$P < 0.05$ vs corresponding treatment (error bars are s.e.m.). Statistical analysis consisted of a two-tailed Student's *t*-test (**b**, **m**, **n**) or a One-way ANOVA followed by Dunnett multiple comparison test (**e**–**k**). Source data are provided as a Source data file.

**Table 1 Morphometric and echocardiographic characteristics of WT versus miR-106b~25 KO mice subjected to sham or TAC surgery for 4 weeks.**

| | Sham | | TAC | |
| | WT | KO | WT | KO |
|---|---|---|---|---|
| $n$ | 10 | 11 | 9 | 10 |
| BW (g) | 21.2 ± 0.2 | 21.3 ± 0.4 | 21.0 ± 0.5 | 21.0 ± 0.4 |
| LV mass (mg) | 107 ± 5 | 80 ± 5* | 145 ± 12* | 110 ± 8# |
| LV mass/BW (mg/g) | 5.1 ± 0.3 | 3.8 ± 0.4* | 7.1 ± 0.7* | 5.3 ± 0.5# |
| IVSd (mm) | 0.83 ± 0.05 | 0.68 ± 0.03* | 1.04 ± 0.06* | 0.89 ± 0.04# |
| IVSs (mm) | 1.16 ± 0.06 | 1.04 ± 0.03 | 1.40 ± 0.07* | 1.16 ± 0.05# |
| LVIDd (mm) | 4.09 ± 0.08 | 4.15 ± 0.09 | 4.10 ± 0.14 | 4.41 ± 0.11 |
| LVIDs (mm) | 2.87 ± 0.08 | 3.14 ± 0.08 | 3.14 ± 0.07 | 3.61 ± 0.15*# |
| LVPWd (mm) | 0.89 ± 0.04 | 0.65 ± 0.04* | 1.04 ± 0.06* | 0.70 ± 0.05*# |
| LVPWs (mm) | 1.11 ± 0.05 | 0.83 ± 0.03* | 1.30 ± 0.08* | 0.88 ± 0.05*# |
| EF (%) | 64 ± 1 | 57 ± 1* | 54 ± 2* | 45 ± 3*# |
| FS (%) | 29 ± 1 | 25 ± 1* | 23 ± 1* | 18 ± 1*# |
| E/A (mm/s) | 1.73 ± 0.11 | 1.53 ± 0.10 | 1.62 ± 0.10 | 1.57 ± 0.24* |

Data are expressed as means ± SEM.
BW body weight, LV left ventricular, IVSd interventricular septal thickness at end-diastole, IVSs interventricular septal thickness at end-systole, LVIDd left ventricular internal dimension at end-diastole,
LVIDs left ventricular internal dimension at end-systole, LVPwd left ventricular posterior wall thickness at end-diastole, LVPws left ventricular posterior wall thickness at end-systole, EF ejection fraction, FS
fractional shortening, E/A Doppler E/A ratio.
*Indicates $P < 0.05$ vs sham group subjected to treatment with a control antagomir.
#Indicates $P < 0.05$ vs experimental group.

the early postnatal developmental period produced cardiac enlargement in the adult heart without classical signs of pathological hypertrophic remodeling.

**The miR-106b~25 cluster stimulates cardiomyocyte proliferation.** To understand how the miR-106b~25 cluster can evoke cardiac growth, we resorted to cardiomyocyte cultures isolated from neonatal hearts, which retain both hypertrophic and proliferative properties[23]. We performed a fluorescence-microscopy-based analysis in neonatal rat cardiomyocytes transfected with either a control precursor miRNA, or precursors for miR-106b, miR-93 or miR-25. At 72 h, cells were stained for sarcomeric α-actinin to distinguish cardiomyocytes from non-myocytes, and we included the proliferation marker 5-ethynyl-2'-deoxyuridine (EdU), a thymidine analog that is incorporated into newly synthesized DNA (Fig. 3a, b). Automated image segmentation and analysis was performed to selectively quantify number of proliferating cardiomyocytes (α-actinin+, EdU+; Fig. 3c) and total number of cardiomyocytes (α-actinin+; Fig. 3d; Supplementary Fig. 4a). The data demonstrate that miR-106b, miR-93, or miR-25 stimulated cardiomyocyte proliferation and cardiomyocyte numbers by a factor of 3 with no substantial differences between the miRNA cluster members.

Next, to evaluate whether the miR-106b~25 cluster would also enhance cardiomyocyte proliferation in vivo, we injected AAV9-miR-106b~25 intraperitoneally in neonatal mice at p1 to elevate cardiac miR-106b~25 cluster expression, administered Edu intraperitoneally at p10 and analyzed the hearts at p12 (Fig. 3e). The hearts of p12 mice injected with AAV9-106b~25 were significantly enlarged compared to those from mice injected with the control AAV9 (Fig. 3f), with no sign of inflammatory cell infiltration or increased cardiac fibrosis content, and no increase in cardiomyocyte size (Fig. 3f, g). Confocal microscopy indicated that the number of both cardiomyocytes in S phase of the cell cycle (α-actinin+, EdU+; Fig. 3h, i; Supplementary Fig. 1e) and mitotic cardiomyocytes (α-actinin+, PH3+; Fig. 3h, j; α-actinin+, Aurora B+; Supplementary Fig. 4b) was significantly increased in hearts of animals injected with AAV9-miR-106b~25 compared to hearts of animals injected with the control AAV9-MCS.

Next, to ascertain if the individual cluster members showed differences in activating cardiomyocyte proliferation, we also generated AAV9 vectors expressing either miR-106b, miR-93, or miR-25 and administered the viral vectors intraperitoneally in neonatal mice at p1, administered EdU at p10 and analyzed the hearts at p12 (Supplementary Fig. 5a). In each case, mice with increased expression of either miR-106b, miR-93, or miR-25 were significantly enlarged (Supplementary Fig. 5b), showed no increase in cardiomyocyte cell size (Supplementary Fig. 5b, c), demonstrated an increased number of cardiomyocytes in S phase of the cell cycle (α-actinin+, EdU+; Supplementary Fig. 5d, e) and mitotic cardiomyocytes (α-actinin+, PH3+; Supplementary Fig. 5f, g; or α-actinin+, Aurora B+; Supplementary Fig. 5h).

Next, we employed stereology to simultaneously assess left ventricular volumes, CM volumes, CM nuclei densities, CM nucleation, and CM proliferation as described previously[24–26] in wild-type neonatal mice, wild-type mice injected with AAV9-miR-106b~25, and miR-106b~25 knockout mice from birth (p0) to p12, since CM proliferation is still abundant in early postnatal life in the mouse and essentially absent after p15[24] (Fig. 4a). CM nuclei in tissue sections were stained for the cardiomyocyte nuclear marker pericentriolar material 1 (PCM-1), CM multi-nucleation was determined by co-staining with wheat germ agglutinin (WGA) to delineate cell boundaries and PCM-1 to mark CM nuclei and CM proliferation assessed by co-labeling sections with PCM-1 and EdU (Fig. 4b). LV volumes increased from $5.5 ± 0.4\ mm^3$ to $13.2 ± 1.0\ mm^3$ at p6 and p12, respectively, with a tendency for mice that received AAV9-miR-106b~25 to have a slightly higher LV volume (Fig. 4c). The ratio of mono- to binucleated cardiomyocytes changed substantially during the first 12 postnatal days. On p6, a minority of cardiomyocytes (14.1 ± 1.1%) were binucleated, whereas at p12 the majority of cardiomyocytes became binucleated (66.1 ± 1.1%), with mice that received AAV9-miR-106b~25 having a slight reduction in the percentage of binucleated cardiomyocytes at p12 (61.2 ± 2.2%; Fig. 4d). The total number of cardiomyocytes at p12 was considerably larger in mice that received AAV9-miR-106b~25 as were the number of EdU+ cardiomyocytes at p6 and p12 (Fig. 4e, f). In addition, the number of EdU+ cardiomyocytes was lower at p12 in miR-106b~25 knockout mice, indicating a requirement of the endogenous miR-106b~25 cluster for cardiomyocyte proliferation in juvenile hearts (Fig. 4f).

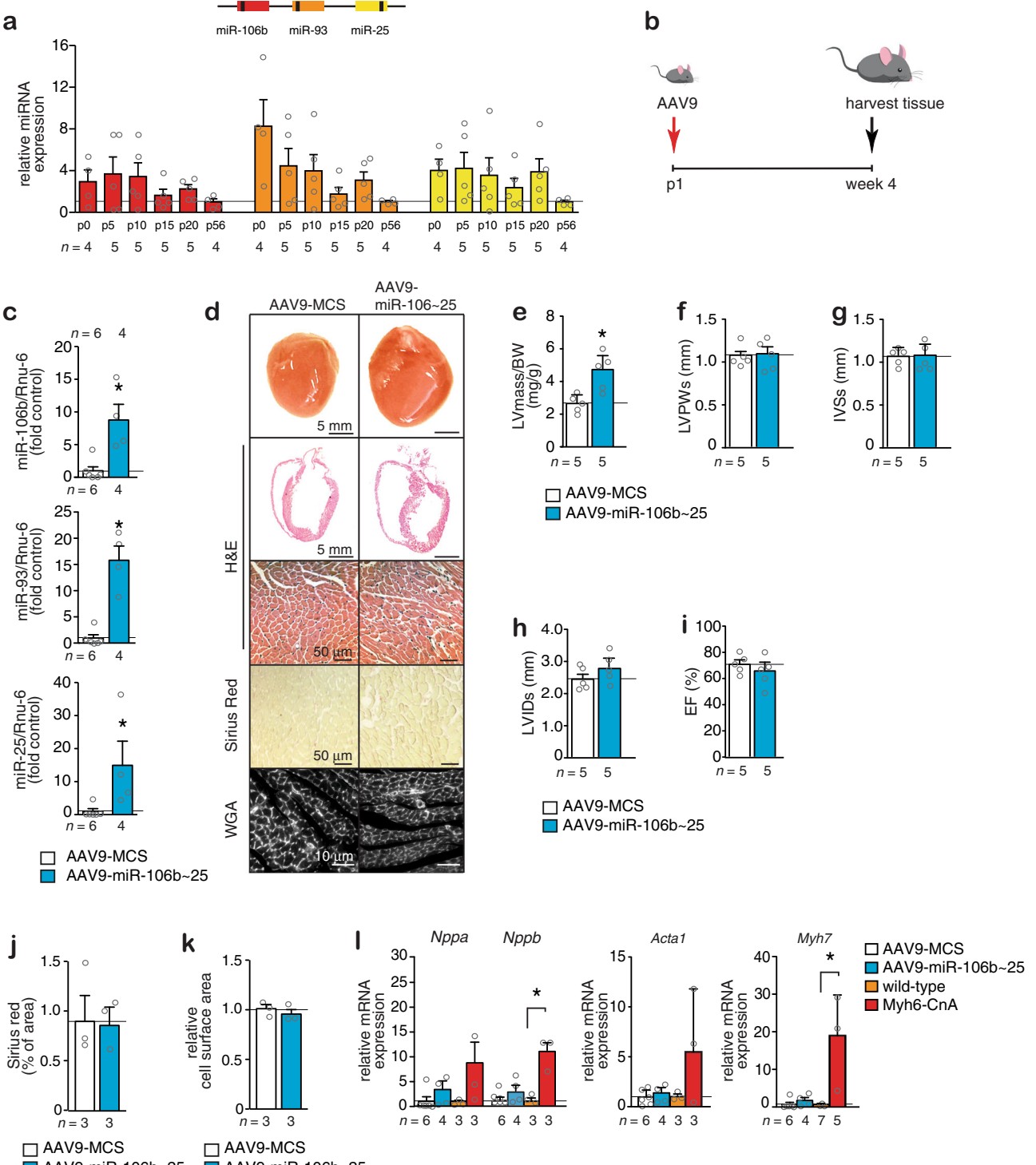

**Fig. 2 Overexpression of the *miR-106b~25* cluster induces cardiac growth with sustained function. a** Real-time PCR analysis of *miR-106b*, *miR-93*, and *miR-25* abundance in mouse hearts at postnatal day 0 (p0), p5, p10, p15, p20, and p56 (2 months) of age. **b** Design of the study. **c** Real-time PCR analysis of *miR-106b*, *miR-93*, and *miR-25* abundance in AAV9-MCS versus AAV9-miR-106b~25 hearts, 4 weeks after injection. **d** Representative images of whole hearts (top panels), H&E-stained sections of four-chamber view (second panel), high magnification H&E sections (third panel), Sirius Red stained sections (fourth panel), and WGA-stained (fifth panel) sections. Quantification of **e** LV/BW ratio, **f** LVPWs, **g** IVSs, **h** LVIDs, and **i** EF of mice that received AAV9-MCS or AAV9-miR-106b~25. Quantification of **j** the fibrotic area by Sirius Red staining and **k** the cell surface areas by WGA-staining. **l** Real-time PCR analysis of *Nppa*, *Nppb*, *Acta1*, and *Myh7* in hearts from mice that received AAV9-MCS or AAV9-miR-106b~25; or WT or calcineurin transgenic (Myh6-CnA) mice, a mouse model for heart failure, *n* refers to number of animals. *$P < 0.05$ vs corresponding control group (error bars are s.e.m.). Statistical analysis consisted of a two-tailed Student's *t*-test (**c**, **e**–**k**) or a One-way ANOVA followed by Dunnett multiple comparison test (**l**). Source data are provided as a Source data file.

**Table 2 Morphometric and echocardiographic characteristics of mice treated for 4 weeks with AAV9-MCS or AAV9-miR-106b~25.**

|  | AAV9-MCS | AAV9-miR-106b-25 |
|---|---|---|
| $n$ | 5 | 5 |
| BW (g) | 29.4 ± 0.5 | 24.3 ± 1.7 |
| LV mass (mg) | 79 ± 7 | 121 ± 26* |
| LV mass/BW (mg/g) | 2.7 ± 0.2 | 4.8 ± 0.6* |
| IVSd (mm) | 0.68 ± 0.03 | 0.91 ± 0.08* |
| IVSs (mm) | 1.05 ± 0.04 | 1.28 ± 0.09 |
| LVIDd (mm) | 3.75 ± 0.12 | 4.08 ± 0.08* |
| LVIDs (mm) | 2.50 ± 0.14 | 2.78 ± 0.19 |
| LVPWd (mm) | 0.84 ± 0.09 | 0.85 ± 0.08 |
| LVPWs (mm) | 1.07 ± 0.06 | 1.09 ± 0.07 |
| EF (%) | 70 ± 3 | 66 ± 5 |
| FS (%) | 33 ± 2 | 31 ± 3 |

Data are expressed as means ± SEM.
*BW* body weight, *LV* left ventricular, *IVSd* interventricular septal thickness at end-diastole, *IVSs* interventricular septal thickness at end-systole, *LVIDd* left ventricular internal dimension at end-diastole, *LVIDs* left ventricular internal dimension at end-systole, *LVPwd* left ventricular posterior wall thickness at end-diastole, *LVPws* left ventricular posterior wall thickness at end-systole, *EF* ejection fraction, *FS* fractional shortening, *E/A* Doppler E/A ratio.
*Indicates $P < 0.05$ vs sham group subjected to treatment with a control antagomir.
#Indicates $P < 0.05$ vs experimental group.

Taken together, these results show that elevated cardiac expression of the complete *miR-106b~25* cluster or the individual cluster members significantly enhance proliferation of at least a subset of cardiomyocytes in vivo.

**The *miR-106b~25* cluster suppress cell cycle inhibitors**. To elucidate the molecular mechanisms underlying the proliferative effects of this microRNA cluster, we performed RNA-seq to assess the transcriptome changes in neonatal rat cardiomyocyte RNA after transfection with *miR-106b*, *miR-93*, or *miR-25* mimics. This analysis identified 1082 genes for *miR-106b*, 1347 genes for *miR-93*, and 1673 genes for *miR-25* upregulated and 1253 genes for *miR-106b*, 1594 genes for *miR-93*, and 2327 genes for *miR-25* downregulated (at 1.0 reads per kilobase of exon model per million mapped reads (RPKM) cutoff and 1.30 fold-change cutoff; Fig. 5a, Supplementary Table S1). We then imposed bioinformatic predictions of miRNA seed sequence interactions with rat transcripts that were downregulated by the miRNAs upon transfection to cardiomyocytes according to the transcriptomic data, yielding 112, 217, and 420 *miR-106b*, *miR-93*, and *miR-25* target genes respectively, with substantial overlap between the *miR-106b* and *miR-93* targetome (Fig. 5b). Next, a bioinformatic protein-protein interaction network was derived that integrates and scores protein interactions across different evidence channels (conserved neighborhood, co-occurrence, fusion, co-expression, experiments, databases, and text mining). For each miRNA in the *miR106b~25* cluster, a subnetwork was extracted. Analysis of the networks showed a particularly dense and often overlapping enrichment for genes functioning in cell cycle regulation, and to a lesser extent networks of genes involved in actin cytoskeletal organization, oxidative stress, and components of the Hippo pathway (Fig. 5c).

*miR-106b* and *miR-93* had overlapping targets among the cyclins and cyclin-dependent kinases including Ccnb1 (cyclin B), Ccna2 (cyclin A), Ccnd1 and Ccnd2 (cyclin Ds) as well as Ccne2 (cyclin E), suggesting that these two miRNAs affected the G2-phase of the cell cycle (Fig. 5d). *miR-25* showed an upregulation of Ccnd1 and Ccnd2 (cyclin Ds) accompanied by a down-regulation of Ccne1 and Ccne2 (cyclin E), Ccna2 (cyclin A) and Ccnb1 (cyclin B), suggesting that this miRNA affected the S-phase of the cell cycle (Fig. 5d). Apart from regulation of specific

Cyclin/Cdk complexes, common targets among all members of the *miR-106b~25* cluster included various cell cycle inhibitors that act on various cell cycle phases, including Cdkn1a (cyclin-dependent kinase inhibitor 1a or p21$^{CIP1}$), Cdkn1c (cyclin-dependent kinase inhibitor 1c or p57$^{KIP2}$), the retinoblastoma transcriptional corepressor 1 (Rb1) inhibitor E2F5, and the G2 checkpoint kinase Wee1 (Fig. 5c, d).

We independently validated our results with an unbiased high-content screen in neonatal rat cardiomyocytes transfected with siRNAs against 18 miR-106b~25 targetome members to screen for their individual contribution to cardiomyocyte proliferation. The data demonstrate that treatment siRNAs against individual targetome members induced only a partial increase in CM proliferation compared to that observed with *miR-106b~25* overexpression, indicating that the effect of the miRNA cluster probably results from a cumulative effect on multiple, cellular mRNA targets (Fig. 5e, f). Finally, we identified perfect matches for the heptametrical seed sequence for the miRNAs in the 3′ UTRs of E2F5 and Cdkn1c that showed evolutionary conservation among vertebrates (Supplementary Fig. 6a, c), validated the target genes using luciferase reporters harboring either the wild-type or site-directed mutagenesis of key nucleotides in the miRNA binding sites (Supplementary Fig. 6b, d), and finally demonstrated derepression of E2F5, Cdkn1c, Wee1 and Ccne1 protein expression in cultured cardiomyocytes transfected with an antimiR for *miR-106b*, *miR-25* or in hearts from *miR-106b~25* null mice by western blotting (Supplementary Fig. 6e–g).

Collectively, these data demonstrate that the *miR-106b~25* cluster regulates densely overlapping networks of genes involved in cell cycle regulation and a number of key cell cycle inhibitors, explaining the proliferative effects of this miRNA cluster.

**The *miR-106b~25* cluster stimulates post-infarction cardiac regeneration**. The adult heart is characterized by a very poor regenerative potential. Given that the *miR-106b~25* cluster is higher expressed in the early postnatal phase and regulates cell cycle regulators and postnatal cardiomyocyte proliferation, we hypothesized that viral delivery of the miRNA cluster could potentially enhance post-infarction regeneration in the adult heart. To test this, adult CD1 mice underwent permanent ligation of left anterior descending (LAD) coronary artery to induce myocardial infarction (MI) and hearts were injected in the peri-infarcted area with AAV9-miR-106~25 or a control AAV9 vector (AAV-MCS) (Fig. 6a). This approach resulted in efficient over-expression of each miRNA over control expression levels after MI (Fig. 6b). At 3 weeks of MI, cross-sectioning hearts from ligation to base demonstrated that hearts injected with the control AAV9 vector displayed the typical large and thinned scarred infarct accompanied with severe biventricular dilation (Fig. 6c, d). In sharp contrast, hearts injected with AAV9-miR-106~25 demonstrated significantly reduced infarct size and preservation of viable LV tissue and cardiac geometry (Fig. 6c, d). The echocardiographic analysis demonstrated near complete normalization of LV mass (Table 3), LVPWd (Fig. 6e), LVIDs (Fig. 6f), LVEF (Fig. 6g) and other functional parameters (Table 3), and a reduction in "stress" marker genes (Supplementary Fig. 6h). Confocal microscopy revealed that mice that received AAV9-miR-106b~25 at three weeks displayed a significant number of EdU-positive cardiomyocyte nuclei in the infarct zone (Fig. 6h) with well-integrated cardiomyocytes within the myocardial structure indicative of active proliferation and regeneration following infarction.

Finally, we cross-bred tamoxifen-inducible genetic lineage-tracing Myh6-MerCreMer mice to a Rosa26 tdTomato reporter mice to permanently mark Myh6-expressing cells (i.e.,

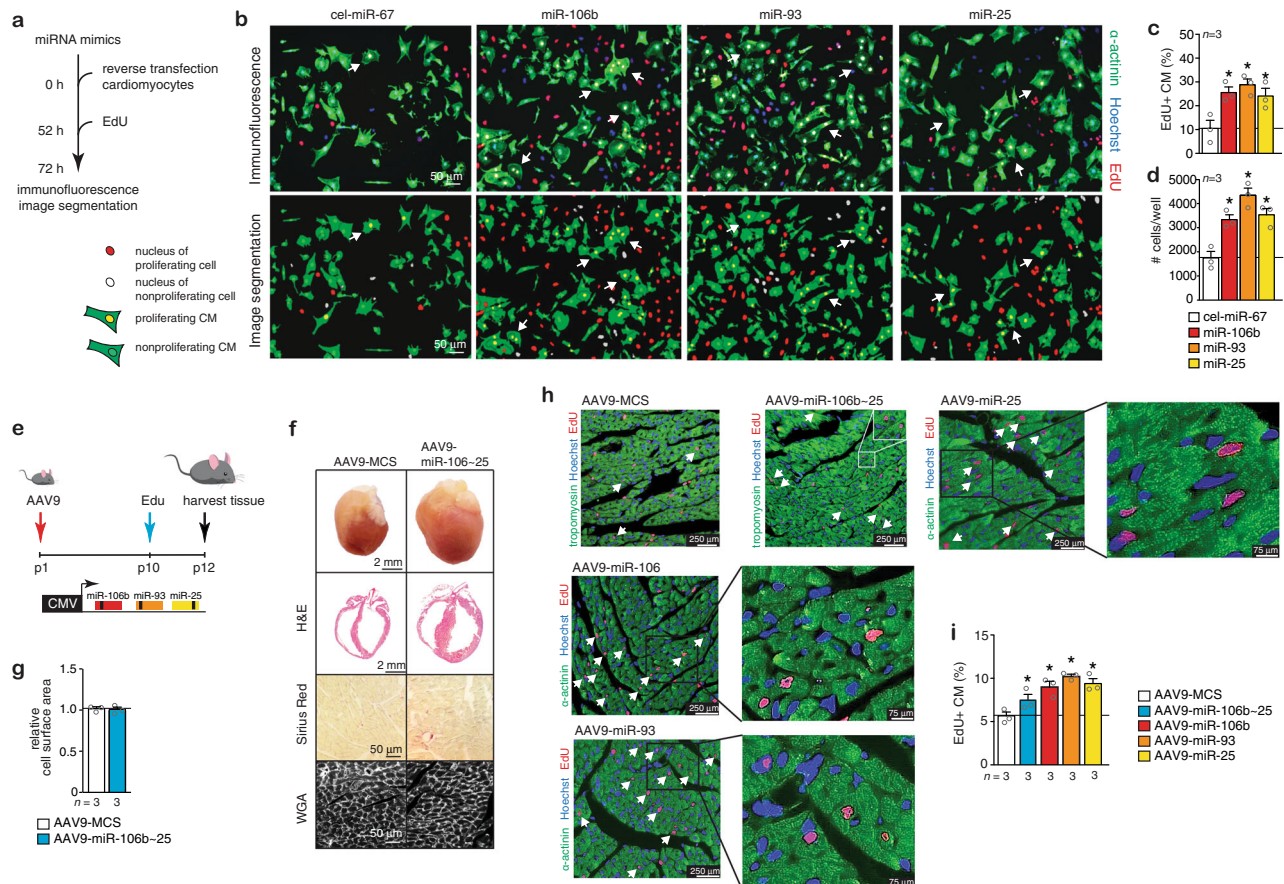

**Fig. 3 Overexpression of the *miR-106b~25* cluster induces cardiomyocyte (CM) proliferation. a** Workflow of the experiment. **b** Representative fluorescent microscopic images of rat CMs transfected with a control precursor miRNA, or precursors for *miR-106b*, *miR-93*, or *miR-25* and stained for α-actinin, 5-ethynyl-2'-deoxyuridine (EdU), and Hoechst. Quantification of **c** the number of proliferating CMs (α-actinin+, EdU+) and **d** total number of CMs (α-actinin+) after transfection with a control miRNA, or *miR-106b*, *miR-93*, or *miR-25*, *n* refers to the number of transfection experiments. **e** Workflow of the study. **f** Representative images of whole hearts (top panels), H&E-stained sections in four-chamber view (second panel), Sirius Red stained sections (third panel) and WGA-stained (fourth panel) sections. **g** Quantification of the cell surface areas from WGA-stained sections. **h** Confocal microscopy images of heart sections of mice treated with AAV9-MCS or AAV9-miR-106b~25 and stained for α-actinin, EdU and Hoechst or α-actinin, pH3 and Hoechst. Quantification of **i** the number of proliferating CMs (α-actinin+, EdU+) and **j** the number of phospho-histone 3 (pH3) positive CMs (α-actinin+, pH3+), *n* refers to number of hearts. *$P < 0.05$ vs corresponding control group (error bars are s.e.m.). Statistical analysis consisted of a two-tailed Student's *t*-test (**g**) or a One-way ANOVA followed by Dunnett multiple comparison test (**c**, **d**, **i**). Source data are provided as a Source data file.

cardiomyocytes) and follow the fate of their cellular descendants in vivo. Accordingly, adult Myh6-MCMR26tdTomato animals were treated with tamoxifen daily for 5 days to label cardiomyocytes with tdTomato. Next, Myh6-MCMR26tdTomato mice underwent permanent ligation of left anterior descending (LAD) coronary artery to induce MI and hearts were injected in the peri-infarcted area with AAV9-miR-106~25 or a control AAV9 vector. A week before sacrifice, all animals received Edu to mark nuclei that are in S phase of the cell cycle and we analyzed EdU+/tdTomato+ cells in tissue sections (Fig. 6i). As expected, tdTomato was exclusively expressed in cardiomyocytes. Importantly, the number of EdU+/tdTomato+ cells in post-infarcted hearts that received AAV9-miR-106~25 was doubled compared to post-infarcted hearts that received a control AAV9 vector. This genetic lineage tracing approach confirms that the AAV9-miR-106~25 vector stimulated the proliferation of pre-existing cardiomyocytes (Fig. 6i–k).

Conclusively, the *miR-106b~25* cluster, relatively high expressed in the early postnatal myocardium that still retains regenerative potential, directs networks of cell cycle regulators and stimulates proliferation of at least a subset of cardiomyocytes in vivo. In adulthood, the relative low cardiac expression of *miR-*

*106b~25* sustains derepression of prohypertrophic cardiomyocyte gene programs that facilitate adverse remodeling in response to overload (Fig. 6l). Exploiting this endogenous regulator between cardiomyocyte hyperplasia and hypertrophy by viral gene delivery enhances the endogenous regenerative capacity of the mammalian myocardium.

## Discussion

Upon after birth, cardiomyocytes enter cell cycle arrest and become terminally differentiated accompanied by polyploidy and hypertrophy as the default growth response to overload or injury[7,27,28]. This terminally differentiated phenotype and reduced cellular plasticity makes the heart more vulnerable in situations when increased workload is required as it either triggers irreversible cell death or hypertrophy[29], which often precipitates in heart failure, a serious clinical disorder that represents the primary cause of morbidity and hospitalization in Western societies.

Here we report on the evolutionarily conserved microRNA cluster that is highly expressed in the early postnatal myocardium and repressed in the adult heart in man and mouse under disease conditions. Remarkably, *miR-106b~25* deficient mice as well as

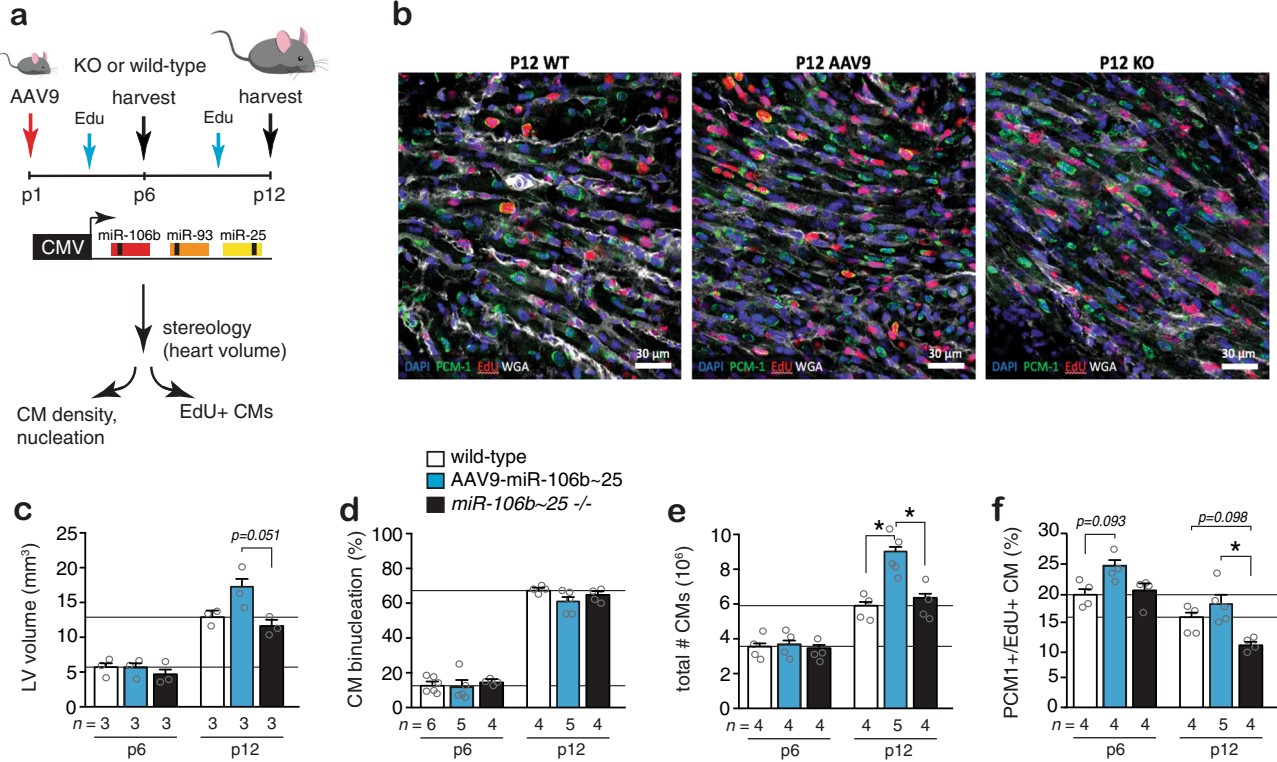

**Fig. 4 The *miR-106b-25* cluster controls the total number of CMs. a** Workflow of the study. **b** Representative confocal images of sections co-stained for pericentriolar material 1 (PCM-1), 4′,6-diamidino-2-phenylindole (DAPI), EdU, and WGA to assess PCM-1+EdU+ CMs in the experimental groups at p12. Quantification of **c** LV volumes, **d** percentage CM binucleation, **e** total number of CMs and **f** proliferating (PCM-1+, EdU+) CMs at p6 and p12 in the experimental groups, *n* refers to number of hearts. *$P < 0.05$ vs corresponding control group (error bars are s.e.m.). Statistical analysis consisted of a One-way ANOVA followed by Dunnett multiple comparison test (**c–f**). Source data are provided as a Source data file.

mice receiving antagomirs for either *miR-106b*, *miR-93*, or *miR-25*, display spontaneous cardiomyocyte hypertrophy and eccentric remodeling, mechanistically explained by the derepression of prohypertrophic downstream targets, most notably the bHLH transcription factor *Hand2*[20], as well as myocyte enhancer factor-2d (*Mef2d*), which serves as a terminal branch of stress signaling pathways that drive pathological cardiac remodeling[30].

Overexpression of the *miR-106b~25* cluster, or the individual cluster members *miR-106b*, *miR-93*, or *miR-25*, by adeno-associated viral (AAV) vectors, stimulated cardiomyocyte proliferation, at least in a subset of cardiomyocytes, by targeting a network of genes with cell cycle regulatory functions including the key cell cycle inhibitors *E2f5*, *Cdkn1c*, *Ccne1*, and *Wee1*, positive cell cycle regulators that are abundantly expressed in the fetal and neonatal heart[31,32]. In the adult heart, cyclin-dependent kinase inhibitors, negative regulators of the cell cycle, are more prevalent[31,32]. In line, forced overexpression of cyclin D2, a positive regulator of the G1/S transition, induced DNA synthesis, and proliferation in mammalian cardiomyocytes[31,32]. Additionally, overexpression of cyclin A2, which promotes the G1/S and G2/M transitions, results in cardiomyocyte proliferation[33], improved cardiac function after ischemic injury in mice[34] and pigs[35].

That members of the *miR-106b~25* cluster can evoke cardiomyocyte proliferation is confirmed by an unbiased, high-content screen to identify proliferative microRNAs[36], while more recently, *miR-25* was demonstrated to provoke cardiomyocyte proliferation in zebrafish by repressing the cell cycle inhibitor *Cdknc1* and tumor suppressor *Lats2*[37]. Our results also revealed components of the Hippo/Yap pathway as *miR-106b~25* targetome members. Hippo signaling has been widely studied in the context of cardiac

regeneration[38,39]. In line, embryonic overexpression of Yap in mice induces hyperproliferation of cardiomyocytes and severely disproportional ventricles and death[38,40,41], while forced expression of Yap in the adult heart provokes cardiomyocyte cell cycle re-entry and regeneration postinfarction injury[40,42]. However, unrestrained Yap activation may also display unwanted effects in pressure overloaded hearts due to cardiomyocyte dedifferentiation[43].

Interestingly, contradicting effects of *miR-25* in the rodent heart have been reported. Some reports indicate that inhibition of *miR-25* expression can lead to derepression of the target gene *Serca2a* and improve cardiac function[44], and others report protection against oxidative stress or apoptosis induced by sepsis[45,46]. In contrast, others report that overexpression of *miR-25* is innocuous and induces proliferation by altering cell cycle genes in zebrafish[37], while here we report cardiac enlargement secondary to enhanced cardiomyocyte proliferation, which at first sight could be misinterpreted as a pathological phenotype. From a therapeutic perspective, *miR-25* loss-of-function approaches have also shown disparate results from improving contractility on the one side[44], or inducing high blood pressure[47], atrial fibrillation[48], eccentric remodeling, and dysfunction[20,47], on the other. It should be noted that distinct chemistries of antisense oligonucleotides can show quite different specificity or even cause side-effects that may explain the opposing observations[49,50]. To avoid the uncertainty surrounding the use of oligonucleotide chemistries, here we resorted to an unequivocal gene deletion strategy where *miR-106b~25* null mice display pathological cardiomyocyte hypertrophy, fibrosis, cardiac dilation and dysfunction, phenotypes that were recapitulated when silencing the individual cluster members with a 2′Ome antisense chemistry. Using the same gene

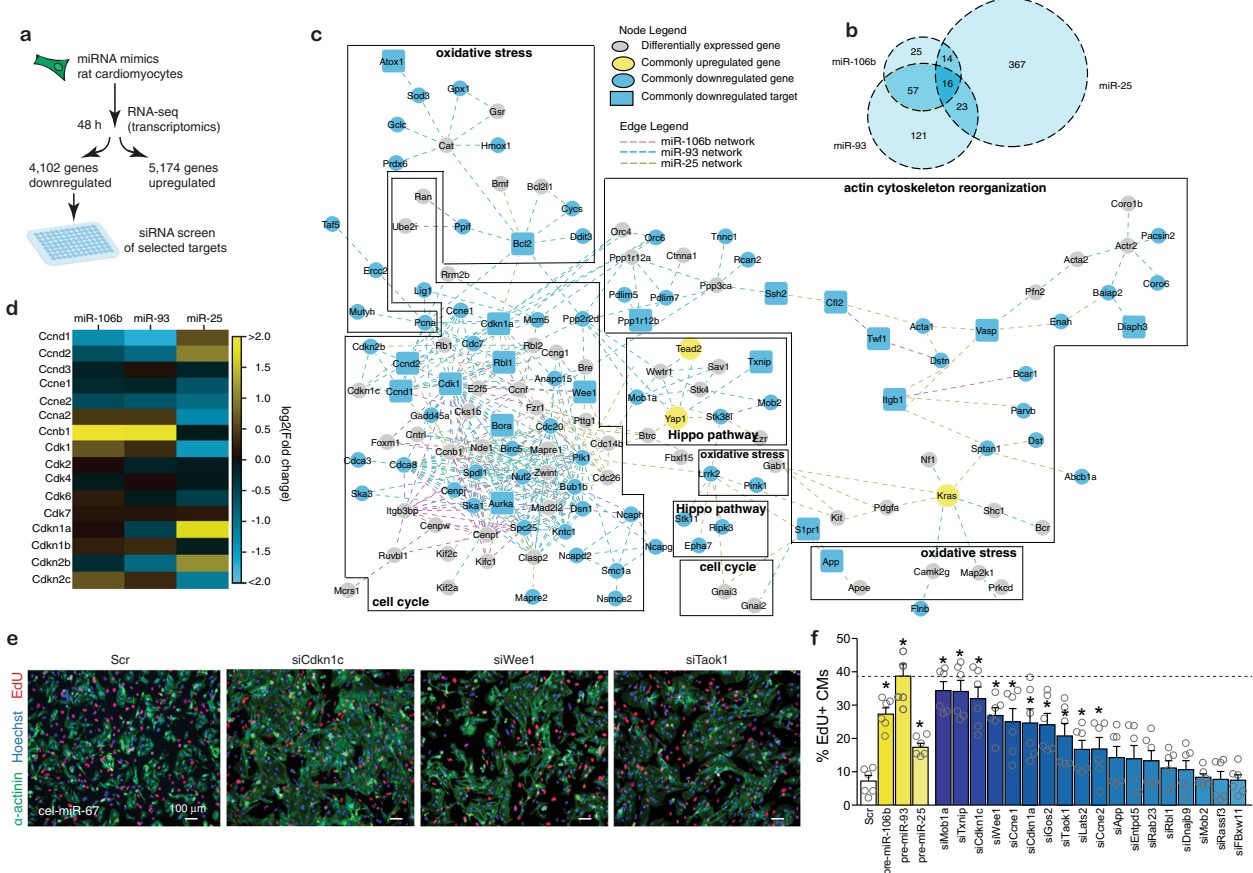

**Fig. 5 The *miR-106b-25* cluster suppresses cell cycle inhibitors. a** Workflow for the experiment and miRNA target validation. **b** Venn diagram showing the relationship between the sets of downregulated genes with *miR-106b, miR-93,* or *miR-25* seed regions and a 1.3-fold change cut-off with respect to *cel-miR-67.* **c** A compartmentalized network of differentially expressed genes involved in cell cycle regulation. Each connection is color-coded according to the miRNA that regulates the differentially expressed gene (magenta for *miR-106b,* cyan for *miR-93* and dark yellow for *miR-25*). Genes indicated by blue circles are downregulated by at least one miRNA while downregulated genes shown in blue squares are found to be bioinformatic targets by at least 1 miRNA. Genes represented by yellow circles are upregulated genes by all three miRNAs. **d** Heatmap representation of cell cycle regulators differentially expressed by *miR-106b~25* relative to *cel-miR-67.* **e** EdU staining of rat CMs treated with selected siRNAs. **f** Percentage of proliferating rat CMs after siRNA treatment, mean ± s.e.m. represents *n* = 6 biologically independent wells examined over one independent experiment. *P < 0.05 vs corresponding control group (error bars are s.e.m.). Statistical analysis consisted of a two-tailed Student's *t*-test (**f**). Source data are provided as a Source data file.

deletion approach, *miR-106b~25* knockout mice show enhanced paroxysmal atrial fibrillation related to disruption of a paired-like homeodomain transcription factor 2 homeobox gene (*Pitx2*) driven mechanism that controls the expression of the *miR-17~92* and miR-106b~25 clusters[51]. In line, *Pitx2* lies in close proximity to a major atrial fibrillation susceptibility locus on human chromosome 4q25 identified in genome-wide association studies[52]. Taken together, exceptional scrutiny should be considered when designing silencing strategies to therapeutically intervene in *miR-25* expression in heart disease.

The combined observations in this study suggest a model whereby defined orchestration of cell cycle regulators underlies the developmental cell cycle arrest of postnatal cardiomyocytes. Moreover, the characteristics of *miR-106b~25* expression in this developmental time frame and its targetome provides a mechanistic explanation for cell cycle exit toward the acquirement of the terminally differentiated phenotype. Hence, when *miR-106b~25* expression is higher, as is the case in the early postnatal heart, cell cycle inhibitors including *E2f5*, *Cdkn1c*, *Ccne1* and *Wee1* are actively suppressed resulting in a proliferative state and cardiomyocyte hyperplasia, while differentiation programs elicited by *Hand2* and *Mef2d* are suppressed. Vice versa, in the overloaded or injured adult heart when *miR-106b~25*

expression is lower, cell cycle re-entry is actively suppressed by the derepression of cell cycle inhibitors, and a prohypertrophic terminal differentiation program is promoted. Taking advantage of the regulatory function between cardiomyocyte hyperplasia and hypertrophy by viral gene delivery of *miR-106b~25* produced regeneration of the adult myocardium in response to chronic ischemic injury. Our data demonstrate that exploitation of conserved epigenetic molecular programs can enhance the regenerative capacity of the injured myocardium.

## Methods

**Human heart samples**. Approval for studies on human tissue samples was obtained from the Medical Ethics Committee of the University Medical Center Utrecht, The Netherlands, and by the Ethical Committee of the University Hospital Hamburg, Germany (Az. 532/116/9.7.1991). All patients or their relatives gave written informed consent before operation. In this study, we included tissue from the left ventricular free wall of patients with end-stage heart failure secondary to ischemic heart disease. Control tissue was taken from the left ventricular free wall of refused donor hearts. Failing hearts were also obtained from patients undergoing heart transplantation because of terminal heart failure. Non-failing donor hearts that could not be transplanted for technical reasons were used for comparison. The donor patient histories did not reveal any signs of heart disease.

**Mouse models**. Mice homozygous null for the mirc3 cluster (*miR-106b~25*) located in intron 13 of the *Mcm7* (minichromosome maintenance complex

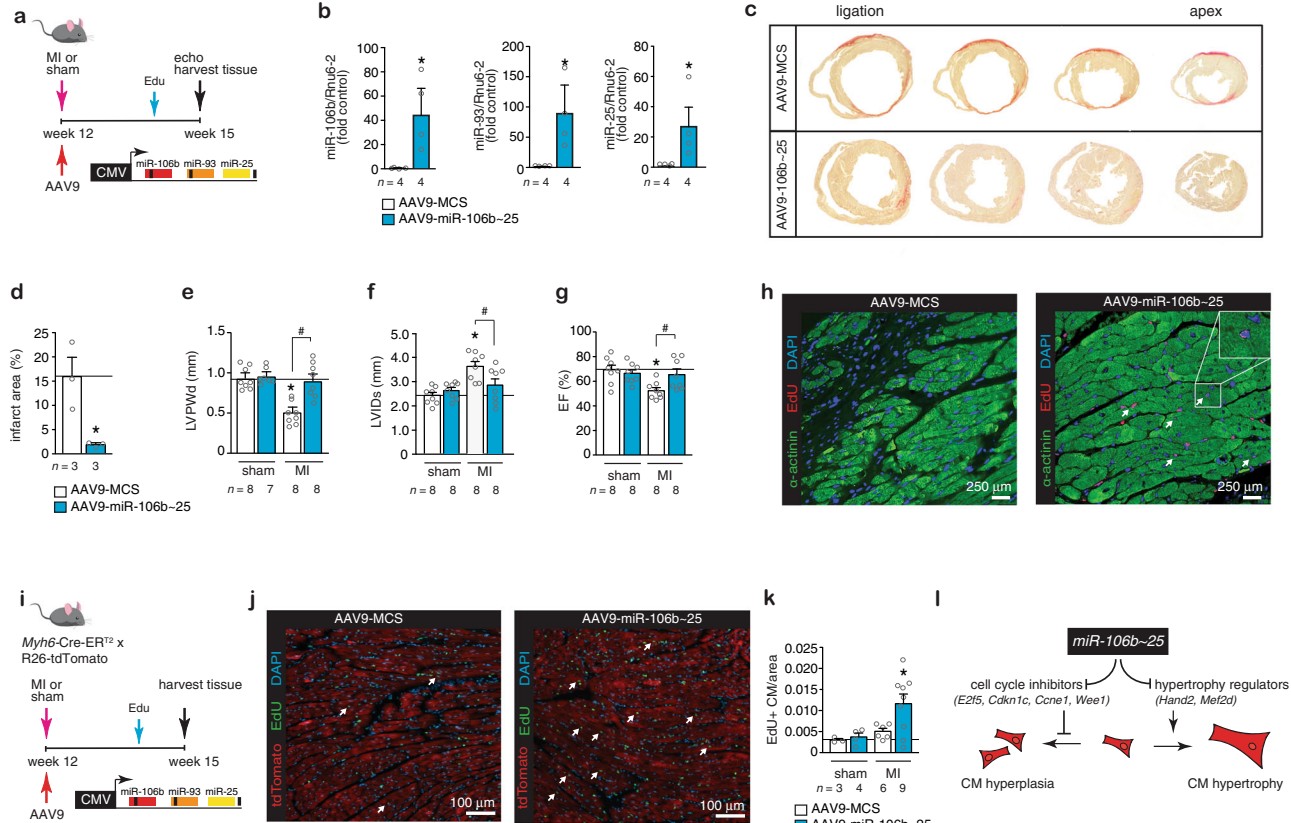

**Fig. 6 The *miR-106b~25* cluster evokes myocardial regeneration. a** Design of the study. **b** Real-time PCR analysis of *miR-106b*, *miR-93* and *miR-25* abundance in infarcted hearts receiving AAV9-MCS or AAV9-miR-106b~25. **c** Representative images of Sirius Red-stained ventricular cross-sections from the point of ligation toward the apex of hearts post-MI, treated with AAV9-MCS or AAV9-miR-106b~25. **d** Quantification of the infarct area of the post-infarcted left ventricles from mice receiving AAV9-MCS or AAV9-miR-106b~25. Quantification of **e** LVPWd, **f** LVIDs, and **g** EF of mice that received AAV9-MCS or AAV9-miR-106b~25 after sham or MI surgery. **i** Representative images of the infarct border zone of mice treated with AAV9-MCS or AAV9-miR-106b~25 after MI surgery, and stained for α-actinin and EdU; *n* refers to the number of animals. **i** Design of the lineage tracing study. **j** TdTomato expression overlaps with 5-ethynyl-2′-deoxyuridine (EdU). **k** Quantification of TdTomato+, EdU+ CMs. **l** Schematic representation of the model. *P < 0.05 vs corresponding control group; #P < 0.05 vs corresponding treatment group (error bars are s.e.m.). Statistical analysis consisted of a two-tailed Student's *t*-test (**b**, **d**) or a One-way ANOVA followed by Dunnett multiple comparison test (**e–g**, **k**). Source data are provided as a Source data file.

**Table 3 Morphometric and echocardiographic characteristics of mice subjected to sham or MI surgery and treated for 3 weeks with AAV9-MCS or AAV9-miR-106b-25.**

| | Sham | | MI | |
|---|---|---|---|---|
| | **AAV9-MCS** | **AAV9- miR-106-25** | **AAV9-MCS** | **AAV9- miR-106-25** |
| *n* | 8 | 8 | 8 | 8 |
| BW (g) | 22.8 ± 2.1 | 25.3 ± 2.0 | 23.8 ± 1.5 | 24.3 ± 1.7 |
| LV mass (mg) | 84 ± 3 | 90 ± 2 | 88 ± 5 | 107 ± 4*# |
| LV mass/BW (mg/g) | 3.2 ± 0.2 | 3.8 ± 0.4 | 5.2 ± 0.4* | 6.4 ± 1.0* |
| IVSd (mm) | 0.79 ± 0.03 | 0.81 ± 0.02 | 0.75 ± 0.03 | 0.90 ± 0.03*# |
| IVSs (mm) | 1.18 ± 0.05 | 1.23 ± 0.07 | 1.10 ± 0.04 | 1.40 ± 0.06*# |
| LVIDd (mm) | 3.62 ± 0.07 | 3.78 ± 0.10 | 4.53 ± 0.19* | 4.00 ± 0.02# |
| LVIDs (mm) | 2.42 ± 0.13 | 2.56 ± 0.14 | 3.55 ± 0.20* | 2.79 ± 0.26# |
| LVPWd (mm) | 0.88 ± 0.03 | 0.86 ± 0.05 | 0.55 ± 0.05* | 0.87 ± 0.10# |
| LVPWs (mm) | 1.16 ± 0.05 | 1.21 ± 0.06 | 0.69 ± 0.07* | 1.05 ± 0.13# |
| EF (%) | 69 ± 4 | 67 ± 3 | 52 ± 2* | 65 ± 4*# |
| FS (%) | 33 ± 3 | 31 ± 2 | 22 ± 1* | 31 ± 3*# |

Data are expressed as means ± SEM.
*BW* body weight, *LV* left ventricular, *IVSd* interventricular septal thickness at end-diastole, *IVSs* interventricular septal thickness at end-systole, *LVIDd* left ventricular internal dimension at end-diastole, *LVIDs* left ventricular internal dimension at end-systole, *LVPwd* left ventricular posterior wall thickness at end-diastole, *LVPws* left ventricular posterior wall thickness at end-systole, *EF* ejection fraction, *FS* fractional shortening, *E/A* Doppler E/A ratio.
*Indicates P < 0.05 vs sham group subjected to treatment with a control antagomir.
#Indicates P < 0.05 vs experimental group.

component 7) gene were generated previously[21], obtained from the Jackson Laboratory (Mirc3tm1.1Tyj/J, Stock No: 008460) and maintained in a B6SV129F1 background. Both male and female *miR-106b~25* null mice of 3–6 months of age were used in this study. Other mice used in this study were 3–6-month-old male calcineurin transgenic mice in a B6SV129F1 background expressing an activated mutant of calcineurin in the postnatal heart under control of the 5.5 kb murine *myh6* promoter (MHC-CnA)[53]; male and female CD1 wild-type mice ranging between postnatal (p) day 0 and p56; male and female B6SV129F1 wild-type mice of 3–6 months of age (Charles River Laboratories). Cre-responsive Rosa26-TdTomato (R26-TdT) reporter mice[54] were crossbred with *Myh6*-mER-Cre-mER mice[55] to generate *Myh6*-mER-Cre-mER R26-TdT mice in a B6SV129F1 background and both male and female mice of 3–6 months of age were used. Tamoxifen dissolved in corn oil was administered prior to myocardial infarction (20 mg/kg). All animal studies were performed in accordance with local institutional guidelines and regulations and were approved by the animal review committee of Medanex Inc., the International Centre for Genetic Engineering and Biotechnology (ICGEB) and the University of Minnesota. Sample size was determined by a power calculation based upon an echocardiographic effect size. Randomization of subjects to experimental groups was based on a single sequence of random assignments. Animal caretakers blinded investigators to group allocation during the experiment and/or when assessing the outcome.

**Production of recombinant AAV vectors**. The precursors of *mmu–miR-106b*, *mmu–miR-93*, and *mmu–miR-25* plus upstream and downstream flanking region sequences (total approximately 200 base pairs) were amplified from mouse genomic DNA isolated from a wild-type CD1 mouse heart, using QIAamp DNA mini kit (Qiagen), according to the manufacturer's instructions. The primers used to amplify the precursor sequences were: forward primer: 5′-GTATCATAAGGA TCCCTTTCCACTGCTCTGGTGAG-3′ and reverse primer: 3′-GTATCATAAG TCGACCTCACCTAGCTGTCTGTCC-5′. The amplified sequences were cloned into the pZac2.1 vector (Gene Therapy Program, Penn Vector core, University of Pennsylvania, USA) using the restriction enzymes BamH I and Sal I. Recombinant AAV serotype 9 vectors were generated at the AAV Vector Unit of ICGEB, Trieste (Italy), as described previously[36]. CD1 mice at postnatal day 1 were intraperitoneal injected with an empty AAV9 vector (AAV9-MCS; multiple cloning site, negative control) or AAV9-miR-106b~25 at a dose of $1 \times 10^{11}$ viral genome particles per animal, using an insulin syringe with 30-gauge needle. 12 days after injection, the hearts were collected for histological analysis.

**Aortic banding, myocardial infarction, AAV9 delivery, and transthoracic echocardiography**. Transverse aortic constriction (TAC) or sham surgery was performed in 2–6 month-old B6SV129F1 mice by subjecting the aorta to a defined 27 gauge constriction between the first and second truncus of the aortic arch as described previously[56,57]. Myocardial infarction (MI) was produced in 2–6 month-old CD1 mice by permanent left anterior descending (LAD) coronary artery ligation[36]. Briefly, mice were anesthetized with an intraperitoneally injection of ketamine and xylazine, endotracheally intubated, and placed on a rodent ventilator. Body temperature was maintained at 37 °C on a heating pad. The beating heart was accessed via a left thoracotomy. After removing the pericardium, a descending branch of the LAD coronary artery was visualized with a stereomicroscope (Leica) and occluded with a nylon suture. Ligation was confirmed by the whitening of a region of the left ventricle, immediately post-ligation. Immediately after MI surgery, adult mice received an intracardiac injection of AAV9 vectors (AAV9-MCS or AAV9-miR-106b~25) at a dose of $1 \times 10^{11}$ viral genome particles per animal. 5-ethynyl-2′-deoxyuridine (EdU, Life Technologies) was administered intraperitoneally (500 μg per animal) every 2 days, for a period of 10 days. For Doppler-echocardiography, mice were shaved and lightly anaesthetized with isoflurane (mean 3% in oxygen) and allowed to breathe spontaneously via a nasal cone. Non-invasive, echocardiographic parameters were measured using a RMV707B (15–45 MHz) scan-head interfaced with a Vevo-770 high frequency ultrasound system (VisualSonics). Long-axis ECG-triggered cine loops of the left ventricular (LV) contraction cycle were obtained in B-mode to assess end-diastolic/systolic volume. Short-axis recordings of the LV contraction cycle were taken in M-mode to assess wall thickness of the anterior/posterior wall at the mid-papillary level. Doppler was used to determine the ratio between early (E) and late (A) ventricular filling velocity (E/A ratio) and to calculate the pressure gradient between the proximal and distal sites of the transverse aortic constriction and only mice with a pressure gradient >50 mm Hg were included. From B-mode recordings, LV length from basis to apex, LV internal diameter in systole (LVIDs) and diastole (LVIDd) were determined. From M-mode recordings, LV posterior wall thickness in systole (LV PWs) and diastole (LV PWd) were determined. LV mass was calculated with the following formula: (0.8*(1.04*(((LVIDd + LV PWd + IVSd)^3)−((LVIDd)^3)) + 0.6); fractional shortening (FS) was calculated with the following formula: (LVIDd−LVIDs)/LVIDd * 100). Ejection fraction (EF) was calculated as ((SV/Vd) * 100) with Vs, systolic volume (3,1416*(LVIDs^3)/6), Vd, diastolic volume (3,1416*(LVIDd^3)/6), and SV, stroke volume (Vd-Vs)[57].

**Fluorescent fluorescence activated cell sorting (FACS)**. Neonatal CD1 mice at age p1 randomly received AAV9-MCS or AAV9-miR106b~25. After 10 days, all

mice we administered a single EdU injection and 2 days later cardiomyocytes were isolated and fixed using 4% PFA. Next, cells were permeabilized using 0.1% Triton-X and incubated for 2 h at room temperature with mouse monoclonal antibody [EA-53] to sarcomeric alpha-actinin (1:100, Abcam ab9465), followed by an incubation of 1 h with goat anti-mouse IgG secondary antibody Alexa Fluor-488 conjugated (1:100, ThermoFisher A-11001). Next, cells were further processed using the Click-IT EdU 647 Imaging kit to reveal EdU incorporation, according to the manufacturer's instructions, and stained with Hoechst 33342 (Life Technologies). Acquisition and analysis was performed on a BD FACSCelesta cell analyser. Analysis was performed using FACSDiva Version 6.1.3.

**Histological analysis and (immunofluorescence) microscopy**. Hearts were arrested in diastole, perfusion fixed with 4% paraformaldehyde/PBS solution, embedded in paraffin and sectioned at 4 μm. Paraffin sections were stained with hematoxylin and eosin (H&E) for routine histological analysis; Sirius Red for the detection of fibrillar collagen; and FITC-labeled rabbit polyclonal antibody against wheat-germ-agglutinin (WGA) to visualize and quantify the myocyte cross-sectional area (1:100, Sigma Aldrich T4144). Cell surface areas and fibrotic areas were determined using ImageJ imaging software (http://rsb.info.nih.gov/ij/). For immunofluorescence, paraffin sections were deparaffinized, rehydrated, and permeabilized with 0.5% Triton X-100/PBS, followed by overnight incubation at 4 °C in 1% BSA with primary antibodies: mouse monoclonal antibody [EA-53] to sarcomeric alpha-actinin (1:100, Abcam ab9465), rabbit polyclonal anti-phospho-Histone H3 (Ser10) (1:100, Sigma-Millipore 06–570), rabbit polyclonal anti-Aurora B (1:100, Abcam ab2254). Next, sections were washed with PBS and incubated for 2 h with goat anti-Mouse IgG Secondary Antibody Alexa Fluor-488 conjugated (1:100, ThermoFisher A-11001), donkey anti-rabbit IgG secondary antibody Alexa Fluor-555 conjugated (1:100, ThermoFisher A32794) or goat anti-mouse IgG secondary antibody Alexa Fluor-647 conjugated (1:100, Thermofisher A32728). For Edu staining sections were processed with the Click-IT EdU 555 Imaging kit to according the manufacturer's instructions. The nuclear counter-staining was performed with Hoechst 33342 (Life Technologies) and slides were then mounted in Vectashield (Vector Labs). Slides were visualized using a Zeiss Axioskop 2Plus with an AxioCamHRc.

**Quantification of cardiomyocytes and binucleation**. Procedures were described in detail previously[24,26]. Hearts were injected with 4 M KCl in the LV for cardioplegia, cut into 1–2 mm pieces and transferred to a peel-a-way embedding mold containing pre-warmed 8% gelatin in DPBS at 37 °C. Tissue was frozen using dry-ice chilled isopentane and stored at −80 °C. Forty μm sections were prepared using a cryostat and placed on positively charged glass slides, allowed to dry for 30 min at room temperature (RT). Cryosections were incubated for 40 min in pre-heated DPBS at 37 °C to remove the gelatin, fixed using 4% formaldehyde in DPBS for 15 min at RT, washed twice in 3% bovine serum albumin (BSA) in DPBS for 5 min and then permeabilized with 1% Triton X-100 for 25 min at RT. Sections were incubated with EdU reaction buffer and washed 5 times in DPBS for 3 min. Blocking was done by incubating the tissue with 4% fetal bovine serum (FBS) in 1% BSA in DPBS for 1 h at RT, prior to incubation with rabbit polyclonal anti-PCM-1 antibody (0.4 mg/μl in blocking solution, Sigma-Aldrich HPA023370) overnight, at 4 °C, followed by incubation with goat anti-rabbit IgG secondary antibody Alexa Fluor-488 conjugated (4 mg/μl in blocking solution, ThermoFisher A32731) for 3 h. Cell membranes were stained and nuclei counterstained by incubating the sections with wheat germ agglutinin (WGA) (50 μg/ml) and 4′,6′-diamino-2-fenil-indol (DAPI) (10 μg/ml) for 1 h at RT, respectively, and mounted in fluoroshield mounting media without DAPI and slides sealed using nail polish. Images of LV fragments were produced in a Leica SP8 STED laser scanning confocal microscope. PCM-1 signal was used to define the beginning and ending of Z-Stack, which varied depending on the penetrance of the PCM-1 antibody. Images were composed of an average number of 40–50 stacks, corresponding to a thickness of 20–25 μm. Five images, from different fragments, were collected per heart. After image acquisition, images were quantified with assistance of FIJI and IMARIS. PCM-1+, EdU+ cells were considered when there was an overlap between PCM-1 (green), EdU (red) and DAPI (blue). Cardiomyocyte density and binucleation were determined for each individual heart by averaging the result obtained in the different fragments. The average number of nuclei per cardiomyocyte was calculated using the formula: average CM nuclei number = 2 × (% binucleation) + 1 × (% mononucleation).

**Western blot analysis**. Whole tissue or cell lysates were produced in 150 mM NaCl, 50 mM Tris-HCl, 5 mM EDTA, 50 mM NaF, 1% Igepal™, 0,05 % SDS, 40 mM β-glycerophosphate, 10 mM Na-pyrophosphate, PhosSTOP- and Protease inhibitor cocktail (Roche Applied Science). Samples were boiled in 4x Leammli buffer, including 2% β-mercaptoethanol, for 5 min at 95 °C. SDS-PAGE and western blotting were performed using the Mini-PROTEAN 3 system (Biorad). Blotted membranes were blocked in 5% BSA/TBS-Tween. Primary antibody labeling was performed overnight at 4 °C at a concentration of 2 μg IgG per 7 mL blocking buffer. Antibodies used included were: rabbit polyclonal antibody anti-p57 (H-91) (1:500, SantaCruz sc-8298), rabbit polyclonal antibody anti-E2f5 (1:500, Abcam ab22855), rabbit polyclonal antibody anti-E2F5 (E-19) (1:500, SantaCruz sc-999),

rabbit monoclonal antibody anti-Cyclin E1 (D7T3U) (1:500, Cell Signaling Technology #20808), rabbit polyclonal antibody anti-Wee1 (1:500, Cell Signaling Technology #4936), rabbit polyclonal antibody anti-Mef2d (1:500, Abcam ab104515), mouse monoclonal anti-GAPDH (1:5000, Millipore, MAB374 clone 6C5), mouse monoclonal anti-alpha-Tubulin (1:5000, Sigma-Aldrich T6074) rabbit polyclonal anti-Histone H3 (1:5000, Cell Signaling Technology 9715S) and the secondary polyclonal swine anti-rabbit immunoglobulins/HRP (1:10,000, DAKO P0399) and polyclonal rabbit anti-mouse immunoglobulins/HRP (1:10,000, DAKO P0161). Secondary HRP conjugated antibodies were applied for 1 h at room temperature. Following antibody incubation, blots were washed for $3 \times 10$ min in TBS-Tween. Images were generated using Supersignal West Dura Extended Duration ECL Substrate (Pierce) and the LAS-3000 documentation system (Fuji-Film Life Science). Stripping was performed with Restore Western blot stripping buffer (Pierce). Output intensities were normalized for loading.

**Quantitative PCR**. Total RNA (1 µg) was extracted using miRNeasy Mini Kit (Qiagen) and applied to either miR-based or mRNA based reverse transcription. For miR-based reverse transcription, total RNA was reverse transcribed using miRCURY LNA Universal cDNA synthesis kit (Exiqon) followed by Real-time PCR using predesigned miRCURY LNA PCR primer sets (Exiqon) and miRCURY LNA SYBR Green master mix, according to the manufacturer's instructions[57]. Expression was normalized to expression levels of 5S rRNA. For mRNA-based reverse transcription, total RNA was reverse transcribed using hexameric random primers. The housekeeping gene ribosomal protein L7 (RPL7) was used for normalization. Fold changes were determined using the $2^{-\Delta\Delta CT}$ method. Real-time PCR primer sequences used in the study are listed in Supplementary Table 1.

**Primary cardiomyocyte cultures and transfections**. Cardiomyocyte cultures were isolated by enzymatic dissociation of 1 day-old neonatal rat hearts and processed for immunofluorescence as described previously[58]. Neonatal cardiomyocytes were seeded on Primaria 384-well plates (for microscopy) or in Primaria 10 cm dishes (for western blotting) and one day later, cells were transfected with mimics (Life Technologies) of hsa-miR-106b-5p, hsa-miR-93-5p, hsa-miR-25-3p or cel-miR-67 as control (25 mM) using (Lipofectamine RNAiMAX, Life Technologies). Twenty-four hours after transfection, culture medium was replaced by fresh medium; and 52 h after plating 5 µM 5-ethynyl-2'-deoxyuridine (EdU, Life Technologies) was added for 20 h. For siRNA transfection, selected siRNAs (Dharmacon) at a final concentration of 50 nM were transfected in cardiomyocytes seeded at $7.5 \times 10^3$ cells per well in collagen-coated black clear-bottom 384-well plates (PerkinElmer)[36]. Cells were fixed at 72 h after plating and processed for immunofluorescence.

**Immunofluorescence and image acquisition of cardiomyocytes**. Cells were fixed with 4% paraformaldehyde for 15 min, permeabilized with 0.5% Triton X-100 in phosphate-buffered saline (PBS) solution for 10 min, followed by 30 min blocking in 1% BSA (Roche). Cells were stained overnight at 4 °C with mouse monoclonal antibody [EA-53] to sarcomeric alpha-actinin (1:100, Abcam ab9465), followed by a goat anti-mouse IgG secondary antibody Alexa Fluor-488 conjugated (1:100, ThermoFisher A-11001). Next, cells were further processed using the Click-IT EdU 555 Imaging kit to reveal EdU incorporation, according to the manufacturer's instructions, and stained with Hoechst 33342 (Life Technologies). Image acquisition was performed using an ImageXpress Micro automated high-content screening fluorescence microscope at 10x magnification; a total of 16 images were acquired per wavelength, well and replicate, corresponding to ~2500 cells analyzed per condition. Image analysis was performed using the 'Multi-Wavelenght Cell Scoring' application module implemented in MetaXpress software (Molecular Devices)[36]. Proliferating cardiomyocytes were identified by a positive signal for the proliferation marker EdU and a positive signal for sarcomeric α-actinin.

**Luciferase-reporter assays**. Constructs bearing 309 bp of murine E2f5 3'UTR (pMIR-E2f5), 591 bp of murine Cdkn1a 3'UTR (pMIR- Cdkn1a), 625 bp of murine Cdkn1c 3'UTR (pMIR- Cdkn1c) and 656 bp of murine Mef2d 3'UTR (pMIR-Mef2d) were subcloned into the pmirGLO Dual-Luciferase miRNA Target Expression Vector (Promega). Correspondent seed-sequence mutated Dual-pMIR-report plasmids were obtained using the QuikChange XL Site-Directed Mutagenesis Kit (Agilent Technologies). Low-passage COS7 cells were grown in DMEM (Invitrogen) supplemented with 10% FCS and seeded ($2.5 \times 10^4$) in 48-well plates and transfected at 50–60% confluence with a total of 100 ng Dual-pMIR-report plasmids using X-tremeGENE 9 DNA Transfection Reagent (Roche), followed by transfection with mimics (Life Technologies) of hsa-miR-106b-5p, hsa-miR-93-5p, hsa-miR-25-3p or cel-miR-67 as control (25 mM) using Oligofectamine (Invitrogen). Firefly and Renilla luciferase activities were measured using the Dual Luciferase Reporter Assay System (Promega), according to the manufacturer's instructions.

**Transcriptomic analysis and clustering of fold change expression**. Deep-sequencing of total RNA isolated from neonatal rat cardiomyocyte cultures was performed 72 h after transfection of mimics (Life Technologies) of hsa-miR-106b-5p, hsa-miR-93-5p, hsa-miR-25-3p, or cel-miR-67 as control (25 mM) by IGA

Technology Services (Italy)[36]. RNA purity, integrity and concentration were determined using an Agilent 2100 Bioanalyzer (Agilent Technologies). Only RNAs with a RIN value > 7 and an rRNA 28 S/18 S ratio > 2 were considered for sample preparation. Two µg of total RNA per sample was sequenced on an Illumina HiSeq2000. Two lanes in 7-plex were run obtaining 2 millions of single-reads per sample, 50-bp long. Real-time image analysis, base calling, de-multiplexing, and production of FASTQ sequence files were performed on the HiSeq2000 instrument using the HiSeq Software. Raw sequence files were quality checked using FASTQC software (www.bioinformatics.babraham.ac.uk/projects/fastqc) and trimmed to remove Illumina adaptor using Cutadapt software. The raw sequencing reads were then mapped to Ensembl Rattus norvegicus reference genome (GCA_000001895.4 Rnor 6.0.89.6)[59] using STAR software. Rounded gene counts were normalized to RPKM (reads per kilobase of exon model per million mapped reads) using the rpkm function in the Bioconductor package edgeR[60]. Genes with RPKM values greater than 2.00 in both miRNA and cel-miR-67 transfected rat CMs were considered as expressed genes. Fold changes were taken with respect to the expression upon cel-miR-67 transfection. Genes whose expression fold change were greater than 1.3 were considered as differentially expressed. The complete RNA-seq data sets from this study were deposited at the Gene Expression Omnibus (GEO) with accession number GSE178867.

**Bioinformatics analyses: clustering of fold change expression; pathway enrichment analysis; target predictions; network analyses**. The Pearson correlation between the log2-fold changes for all pairs of miRNA were calculated. Clustering was performed hierarchically using the average linkage criterion with a Euclidean distance metric as implemented in SciPy v0.18.1 (http://www.scipy.org). A dendrogram was then generated using the SciPy clustering package to visualize the arrangement of the resulting cluster. Statistically enriched pathways from the set of differentially expressed genes were determined using a hypergeometric distribution-based statistical method as implemented in the Database for Annotation, Visualization and Integrated Discovery (DAVID) Bioinformatics Resources 6.8. The calculated P-values were then corrected according to Benjamini-Hochberg procedure to control the false discovery rates arising from multiple testing. KEGG pathways with $P < 0.01$ and Benjamini–Hochberg FDR < 0.05 are considered as statistically significant. Since bioinformatic predictions of seed sequence interactions with rat transcripts are not available, we compiled a list of rat miRNA-gene interactions from mouse predictions. Predicted mouse gene targets of the seed sequences (corresponding to miRNA families) of miRNAs that belong to the miR-106b~25 cluster were collected from TargetScanMouse Release 7.1. The scores were calculated to be the most efficient interaction between a mouse gene and a human miRNA in a given miRNA family as determined by the seed sequence. The mouse genes were then translated to its corresponding rat genes through homology using the HomoloGene database. The list of miRNA-gene interactions was filtered to only include genes that were downregulated by the miRNA upon transfection to CMs according to the transcriptomic data (397 downregulated bioinformatic targets by miR-106b, 429 by miR-93 and 358 by miR-25). A bioinformatic protein-protein interaction network was derived from STRING database v10.5, which integrates and scores protein interactions across different evidence channels (conserved neighborhood, co-occurrence, fusion, co-expression, experiments, databases, and text mining) and combines the scores from these channels. Only those interactions solely involving rat proteins were considered. To obtain a high confidence interaction network, a cut-off score of 700 (out of 1000) on the combined score was imposed. For each miRNA, a subnetwork was extracted containing only the differentially expressed genes upon imposing a 1.3-fold-change cut-off and the genes of interest in this study. These subnetworks were then merged into a single multi-network wherein the gene components can now be connected by three interactions, one for each miRNA. To further elucidate the multi-network, we cataloged the set of genes that are involved in biological processes of interest using the EMBL annotations of gene ontologies. These biological processes include cell division (GO:0051301 and GO:0000086) with 212 annotated genes, actin cytoskeleton (GO:0015629, GO:0030036, GO:0031532, and GO:0008154) with 345 annotated genes and oxidative stress response (GO:0006979) with 143 annotated genes. Furthermore, knowing the Hippo signaling pathway in cardiomyocyte proliferation, we also cataloged the set of genes involved in the canonical[61] (19 genes) and non-canonical[62] (21 genes) pathways in which Yap can be phosphorylated. These gene sets were then used to extract a compartmentalized network of genes that are differentially expressed after transfection of the miR-106b~25 cluster whose interactions can lead to cytokinesis. Network analysis was done using the NetworkX v1.11 package in Python while graph visualization was done using Cytoscape v3.4.0.

**Statistics and reproducibility**. Results shown for images or blots were repeated independently at least once with similar results. The results are presented as mean ± standard error of the mean (SEM). Statistical approaches for bioinformatics analyses are described above. All other statistical analyses were performed using Prism software (GraphPad Software Inc.), and consisted of One-way ANOVA followed by Dunnet's multiple comparison test when group differences were detected at the 5% significance level, or Student's t-test when comparing two experimental groups. Differences were considered significant when $P < 0.05$.

**Reporting summary**. Further information on research design is available in the Nature Research Reporting Summary linked to this article.

## Data availability

The data that support the findings in this study are available within the article and its Supplementary information files. Raw and processed RNAseq data generated in this study have been deposited at the GEO database under accession code: GSE178867. Published resources evaluated included HomoloGene NCBI database (https://www.ncbi.nlm.nih.gov/homologene) and STRING (https://string-db.org/). Any remaining raw data will be available from the corresponding author upon reasonable request. Source data are provided with this paper.

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

## Acknowledgements

E.D. is supported by a VENI award 916-150-16 from the Netherlands Organization for Health Research and Development (ZonMW), an EMBO Long-term Fellowship (EMBO ALTF 848-2013) and a FP7 Marie Curie Intra-European Fellowship (Project number 627539). V.S.P. was funded by a fellowship from the FCT/ Ministério da Ciência, Tecnologia e Inovação SFRH/BD/111799/2015. P.D.C.M. is an Established Investigator of the Dutch Heart Foundation. L.D.W. acknowledges support from the *Dutch CardioVascular Alliance* (ARENA-PRIME). L.D.W. was further supported by grant 311549 from the European Research Council (ERC), a VICI award 918-156-47 from the Dutch Research Council and Marie Sklodowska-Curie grant agreement no. 813716 (TRAIN-HEART).

## Author contributions

E.D., A.R., S.O., and L.O. performed real time PCR experiments. A.R., H.A., and S.O. performed western blots. A.R. performed luciferase assays. E.D., C.T., and R.C. performed transcriptome analysis. H.A. and S.Z. performed surgical procedures in mouse models. E.D. performed echocardiography. E.D., A.R., V.S.P., and D.S.W. performed histology in mouse models. E.D. and A.R. analysed data. E.D. and S.M. performed FACS analyses. L.B. performed microscopic imaging. M.H., R.W., L.Z., D.N. J.V.B., and M.G. provided reagents and models. E.D., M.G. S.S., P.D.C.M., and L.D.W. designed the study. E.D. and L.D.W. wrote the manuscript. E.D, P.D.C.M., and L.D.W. acquired funding for the study. E.D. and A.R. contributed equally as joint first authors.

## Competing interests

E.D., M.G., and L.D.W. filed the data in the manuscript for patent protection. P.D.C.M. and L.D.W. are co-founders and stockholders of Mirabilis Therapeutics BV. The remaining authors declare no competing interests.
