## [Peer Review File · Nature Communications]

Reviewers' comments:

Reviewer #1 (Remarks to the Author):

This study by Dirkx et al. reports on a microRNA cluster (containing miRs-106, -93, and -25) that represses key cell cycle inhibitors in cardiomyocytes and thereby enables their proliferation. Downregulation of this cluster after birth in mice contributes to the loss of cardiomyocyte proliferative capacity and the inability to repair lost myocardium in disease states such as myocardial infarction. The authors make a convincing case that this miR cluster can drive the cardiomyocyte cell cycle and this is certainly a very important finding. Still, several points need to be addressed:

While the manuscript is well written, the organization of the data appears not straightforward. The authors start out with a KO phenotype in TAC-treated mice, before they overexpress the cluster in WT mice, then they overexpress the cluster in vitro, then cluster members (separate figure) in vitro, then the cluster back in vivo in WT (this time with Edu), then cluster members in vivo (different figure), targetome back in vitro, then overexpression in MI model. In between there is the postnatal downregulation of the cluster. A more 'conventional' way to present this set of data would likely facilitate access to these data for the readers of this manuscript.

The authors report on two different disease models, each with different manipulation of the miR cluster (KO vs. AAV). To what extent do the authors think that the hypertrophy-related targets studied in the KOs contribute to the overexpression phenotype observed after MI? Conversely, are the smaller hearts in the KOs attributable to diminished CM proliferation?

The KO Sham heart in Fig 1D appears to exhibit a larger myocardial mass than its WT control?

While the authors make a convincing case that the miR cluster can drive the cardiomyocyte cell cycle they do not formally prove that endogenous miR-106/93/25 controls the CM cell cycle in the heart. I suggest to tone down the respective claims in title, abstract and manuscript text.

The images provided for CM proliferation in myocardium appear somewhat limited as they leave some room for interpretation as to the cell type the respective nucleus is attributed to. e.g. what do the three arrows in 3H point to compared to all the other red nuclei? There is a considerable number of yellow nuclei, suggesting an overlay of green (alpha actinin) and red (EdU) - aren't those the proliferating CMs? Also Fig. 6J appears limited in this respect and AAV control and AAV miR should be represented by myocardial regions that are better comparable. Both Fig. 3 and Fig. 6 deserve higher magnification pictures (and possibly thinner confocal imaging sections) that allow for better attribution of the signals to cardiomyocytes and other cell types. Fig S1 E and F deserve better labelling and detailed description in this respect, e.g. upper panels are control, lower are AAV-miR, colour coding and numbering of P1, P2 and P3 is confusing. How many mice were studied? What is the actual percentage of adult cardiomyocytes positive for Edu? Does the 'significantly increased'

refer to the 0.2%, is this the average? How does this compare to the results of the confocal analyses?

The authors should discuss the time course of the downregulation of the miR cluster in relation to the reported loss of proliferation capacity. It appears that 93 and 25 are lost relatively late in this respect and 106b may then be the responsible one? Is the downregulation occurring in CMs as opposed to other cell types?

The Hippo pathway findings should be discussed with respect to the recent literature (Nature paper by J. Martin group etc.) as should the findings by the Mercola group on miR-25.

Minor:

Scale bars are sometimes missing.

Mice are not 'treated with AAV for 4 weeks' but studied 4 weeks after AAV injection.

Reviewer #2 (Remarks to the Author):

De Windt and colleagues investigated the role of the miR-106b~25 cluster in the heart and discovered that it may promote proliferation of cardiomyocytes in the postnatal hearts.

Overall, the findings are interesting and provide novel findings regarding the function of the miR-106b~25 cluster. The paper is generally descriptive and the possibility that the cardiac phenotypes in loss and gain of function models are mediated through mechanisms other than changes in myocyte proliferation cannot be excluded. Underlying molecular mechanisms are not clearly defined.

Specific comments:

1. The model in Figure 1 appears to have fundamental problems. Before the stress is given, the mouse heart already has a significant phenotype. It is unclear whether the phenotype after TAC is due to the function of the miR-106b~25 cluster during development or during pressure overload stress. For example, if the heart has less myocytes before TAC, it alone may exacerbate cardiac dysfunction due to reduced contractility.

2. The cardiac phenotype is poorly investigated in Figure 1. For example, the authors could have investigated the total number of cardiomyocytes in the heart and the extent of cell death. Does upregulation of Mef2D play a significant role in mediating the cardiac phenotype? Is the cardiac phenotype in Figure 1 caused by pathological hypertrophy, the lack of cardiomyocytes proliferation, or other mechanisms?

3. Similarly, no mechanistic experiment has been conducted to elucidate the mechanism by which the heart treated with AAV9 miR-106b, AAV9 miR-93, or AAV9 miR-6 is enlarged. What is the total cell number in the heart? Similarly, what is the effect of AAV9 miR-106b~25 upon the total cell number in the heart?

4. Although the authors have suggested potential targets in Figure 5, no evidence was provided regarding their mechanistic involvement (significance).

5. Again, the analysis of cardiac phenotype is poorly conducted in Figure 6. What is the cell size or the total cell number in the heart? The data shown in Figure 6j is insufficient as evidence of cardiomyocyte proliferation. Can we really say that the rescue of cardiac function by AAV9-miR106b~25 is mediated through cardiomyocyte proliferation?

6. A paper published in Nature (508, 531-535, 2014) reports an opposite function of miR-25 in the heart. The authors could have discussed this issue.

Reviewer #3 (Remarks to the Author):

In this study, the authors show that miR106b~25 family induces cardiac regeneration in the setting of heart failure. A focus in cardiac regeneration has been to induce existing cardiac myocytes to divide and proliferate. Therefore, any strategy that induces proliferation of adult cardiomyocytes in the region damaged by ischemic injury is of great interest in cardiac regenerative medicine. Of particular interest, one of co-authors in this strategy, Mauro Giacca, devised a strategy to identify miRNAs that affect the proliferation of cardiomyocytes in a very elegant manner (Eulalio et al. Nature, 2012).

Dirkx et al. previously showed that increased calcineurin/Nfat signalling and decreased miR-25 expression integrate to re-express the basic helix-loop-helix (bHLH) transcription factor, Hand2 in the

diseased human and mouse myocardium. On the other hand, Wahlquist et al. reported that same miRNA25 targets SERCA2a in the heart(Nature 2014).

Overall, technically the study is well designed and executed. However, there are several critical issues that are of concern in this study.

1. The authors claimed that end-stage HF animal models were used in entire in-vivo experiment. However, there is no molecular evidence to support this claim. Rather, based on the functional characteristics the authors provided in this manuscript seems more close to hypertrophic state. Therefore, it is highly recommended to provide the miRNA profiling including miRNA-106b~25 cluster during cardiac hypertrophy (eg. 1 or 2wks post TAC will be good model). Since expression pattern of many signature molecules are substantially differed in hypertrophy and adverse heart failure these additional studies are important.

2. A major inconsistency in the study is that Dr. Mauro Giacca's group showed in their paper (Eulalio et al, Nature 2012) that miR-25 is one of the miRNAs that DECREASE cardiomyocyte proliferation (Supplmentary figure 2 in previous Nature paper (Eulalio et al., 2012)). The data of Dr Giacca (an author on this study) directly contradict the authors work.

3. All histological data (MT and Picosirius red etc) show that the size of cardiomyocytes from AAV9-miR-106b~25 injected mouse hearts seem much larger than those of control group. It is unclear how authors measure CSA in this paper (Fig 2d and 3f).

4. In Fig.1, the authors show that knockout of the miR-106b~25 cluster leads to mild cardiac hypertrophy and fibrosis at baseline. This data is basically same as the Figure 6 of the author's previous paper (Nature Cell Biology, 2013). In this previous study, they showed that knock-down of miR-25 using an antagomir led to similar cardiac dysfunction at baseline. Thus, the current Fig 1 does not tell anything about the functional significance of miR-106b and miR-93. It would have been much more informative on the role of miR-106b and miR-93, if they performed similar knockdown experiments with individual antagomirs of miR-106b or miR-93. In Fig 2, they observed the gain-of-function effects of the individual miRs. Then, it is not clear why they did not perform similar experiments for the loss-of-function effects of the individual miRs?

5. In page 5, line 4, they wrote “overexpression of the miR clusters to physiological levels”. What do they mean by “physiological level”? Do they mean the level at p0? It is also unclear whether the expression level was indeed restored to the levels seen at p0 or it was much more up-regulated beyond the p0 level. For clarification, Fig 2C is to be revised to include the expression level of individual miR at p1 when the viruses were injected. It is extremely important to discern whether the proliferative effects of the miR are seen when the expression level was merely restored to the level seen at p0 or when it is elevated to a non-physiologically high level.

6. In Fig 3a~d, the authors utilized cardiomyocytes isolated from neonatal hearts. In neonatal hearts, the expression level of each miR is high. Why would they express more miRs on top of the already high level of miR? Once again, this reviewer suspect that the proliferative effects of miRs are

achieved when the miRs are overexpressed beyond the physiological level. To address this issue, the authors should use cardiomyocytes isolated from adult hearts where the expression levels of miR hit bottom.

7. The same logics mentioned above (#3) applies to Fig 3e~j. The virus should be injected to adult mice and the expression level of miR should be carefully monitored. Once again, the important issue is whether the AAV-mediated expression of miR restores the lowered level of miRs to the level seen at p0 or to the level of way over the physiological level.

8. miR-25 is overexpressed in several cancer cells and miR-25 has been reported as a tumor suppressor via SOX4 and CyclinD1, or EZH2 activation especially on G1 phase of the cell cycle. (Chen et al., *Tumour Biol.*, 2017; Xu et al., *Asian Pac J Trop Med*, 2013; Esposito et al., *JCEM*, 2012 etc). On the other hands, authors claim that miR-25 induces proliferating cardiomyocytes. This should be clarified.

9. Other previous studies showed that miR25 is up-regulated in the failing hearts. The authors report the exact opposite observations. Any thoughts or explanations on this discrepancy? Better be mentioned in the discussion section.

Reply to Reviewer #1 (Revision 1)

Dirkx, Raso et al. NCOMMS-17-20364. "A microRNA program controls the transition of cardiomyocyte hyperplasia to hypertrophy and stimulates mammalian cardiac regeneration".

We thank Reviewer #1 for his/her helpful and constructive remarks that improved the manuscript. We have carefully considered all of the suggestions proposed and revised the manuscript accordingly. Please note that all changes in the manuscript are marked in yellow.

Below the Reviewer will find a point-by-point rebuttal to his/her questions (Q).

Major

Q1 **"While the manuscript is well written, the organization of the data appears not straightforward. The authors start out with a KO phenotype in TAC-treated mice, before they overexpress the cluster in WT mice, then they overexpress the cluster in vitro, then cluster members (separate figure) in vitro, then the cluster back in vivo in WT (this time with Edu), then cluster members in vivo (different figure), targetome back in vitro, then overexpression in MI model. In between there is the postnatal downregulation of the cluster. A more 'conventional' way to present this set of data would likely facilitate access to these data for the readers of this manuscript"**

Authors' response Q1: We are grateful for the Reviewer's compliments. We have considered at length whether a more conventional presentation of the data - *starting with (gain- and loss-of-function) cell culture experiments and ending with (gain- and loss-of-function) in vivo mouse studies* - would yield an equally powerful message to the readership.

We are convinced, however, that the main message of the manuscript - *the endogenous miR-106b~25 cluster is higher expressed in the early postnatal myocardium, decreases in expression towards adulthood and orchestrates the transition of cardiomyocyte hyperplasia towards cell cycle arrest and subsequent cardiomyocyte hypertrophic growth by virtue of its targetome* - is best helped by emphasizing the opposite phenotypes *in vitro* and *in vivo* depending on the myocardial expression level of this microRNA cluster.

Accordingly, in **Fig.1** and **Suppl.Fig.1,2** we emphasize the hypertrophic phenotype obtained with a genetic knockout of the miR-106b~25 cluster; i.e. gene-targeted *miR-106b~25* null mice display spontaneous hypertrophic remodeling and exaggerated remodeling to overload by derepression of the prohypertrophic transcription factors Hand2 and Mef2d (**Fig.1**). Additionally, as requested by Reviewer #3, we now also demonstrate that single antagomir-mediated silencing of the individual miR-106b~25 cluster members causes spontaneous hypertrophic remodeling of the murine heart (**Suppl. Fig 2**).

In stark contrast, **Figs 2-6** and **Suppl.Figs 3-6** delineate the findings with gene delivery of miR-106b~25 to the mouse heart where it provokes cardiomyocyte proliferation by targeting a network of negative cell cycle regulators, a situation that naturally occurs in the early postnatal myocardium where cardiomyocytes still contain substantial proliferative capacity.

Q2 "The authors report on two different disease models, each with different manipulation of the miR cluster (KO vs. AAV). To what extent do the authors think that the hypertrophy-related targets studied in the KOs contribute to the overexpression phenotype observed after MI?"

Authors' response Q2: This is an insightful question. The Reviewer asks whether in the overexpression studies of the *miR-106b~25* cluster, repression of the pro-hypertrophic validated targets Hand2 and Mef2d also participate in re-activation of the cardiomyocyte cell cycle. First, loss-of-function studies for Mef2d (Kim et al. J Clin Invest. 2008) or Hand2 (ref. 20, Dirkx et al. Nat Cell Biol. 2013) indicate no evidence for a potential involvement of either factor in proliferation of postnatal cardiomyocytes, but rather confirm their function in cardiac hypertrophy *in vivo*. Secondly, we performed extensive literature searches to obtain any indication whether or not GO terms of "cell cycle activity" or "proliferation" have been assigned to Mef2d and Hand2, but neither have such indication.

As the Reviewer can perceive from new **Fig.5a,e,f**, we have now performed an unbiased high-content screen in neonatal rat cardiomyocytes with siRNAs against 18 targets of the *miR-106b~25* cluster to screen targetome members for their individual contribution to cardiomyocyte proliferation. We have dedicated our resources (3 siRNAs designed against each target for a total of 18 targets in the unbiased screen is a remarkably expensive undertaking) only to those *miR-106b~25* cluster targets for which stronger evidence of involvement in cell cycle re-activation could be expected. The data demonstrate that treatment with siRNAs against targetome members induced only a partial increase in CM proliferation compared to that observed with *miR-106b~25* overexpression, indicating that the effect of the miRNA cluster probably results from a cumulative effect on multiple, cellular mRNA targets.

- Q3** "Conversely, are the smaller hearts in the KOs attributable to diminished CM proliferation?"
- Q4** "The KO Sham heart in Fig 1D appears to exhibit a larger myocardial mass than its WT control"
- Q5** "While the authors make a convincing case that the miR cluster can drive the cardiomyocyte cell cycle they do not formally prove that endogenous miR-106/93/25 controls the CM cell cycle in the heart. I suggest to tone down the respective claims in title, abstract and manuscript text"

Authors' response Q3, Q4, Q5: Your points are well taken.

Q4: We agree that the sham *miR-106b~25* KO heart is slightly bigger in size than its wild-type counterpart. Indeed, the echocardiographic-derived parameters (see **Table 1**) clearly point to a dilated phenotype with thinned walls (**Fig.1f**), reduced ejection fraction (**Fig.1h**) and reduced LV mass, likely due to the thinning of the LV walls (**Table 1**).

Q3, Q5: We felt we could only answer these questions by employing stereology to simultaneously assess left ventricular volumes, cardiomyocyte (CM) volumes, CM nuclei densities, CM nucleation and CM proliferation as described previously (Alkass et al. Cell. 2015; Sampaio-Pinto V et al. Stem Cell Reports. 2018). In addition, we decided to analyze mouse hearts in the early postnatal phase to address the reviewer's valid question whether the endogenous *miR-106b~25* cluster controls CM cell cycle, since CM proliferation is still abundant in early postnatal life in the mouse and is essentially absent after postnatal day 15 (see e.g. Alkass et al. Cell. 2015).

Accordingly, we included 3 experimental groups: (a) wild-type neonatal mice, (b) wild-type neonatal mice injected with AAV9-miR-106b~25 intraperitoneally at p1 to elevate cardiac *miR-106b~25* cluster expression, and (c) *miR-106b~25* cluster knockout mice. We administered EdU intraperitoneally at p1, p2 and p9 in all groups to mark proliferating cells and analyzed hearts at p6 and p12 (**Fig.4a**).

LV volumes, cardiomyocyte volumes and total number of cardiomyocytes were determined as described previously (Alkass et al. Cell. 2015; Sampaio-Pinto V et al. Stem Cell Reports. 2018). Cardiomyocyte nuclei in tissue sections were unequivocally identified by staining for the cardiomyocyte nuclear marker pericentriolar material 1 (PCM-1).

Cardiomyocyte multinucleation was determined by co-staining with wheat germ agglutinin (WGA) to delineate cell boundaries and PCM-1 for cardiomyocyte nuclei. Cardiomyocyte proliferation was assessed by co-labelling cardiac sections with PCM-1 and EdU (**Fig.4b**).

In neonatal mice, LV volumes increased from $5.5 \pm 0.3 \text{ mm}^3$ to $13.2 \pm 0.8 \text{ mm}^3$, with no substantial differences when mice received AAV9-miR-106b~25 or mice that were genetically deficient for the cluster, although there was a tendency for mice that received AAV9-miR-106b~25 to have a slightly higher LV volume (**Fig.4c**).

The ratio of mono- to binucleated cardiomyocytes changed substantially during the first 12 postnatal days. On p6, a minority of cardiomyocytes ($14.1 \pm 1.1\%$) were binucleated, whereas at p12 the majority of cardiomyocytes became binucleated ($66.1 \pm 0.9\%$), with no substantial differences with mice that received AAV9-miR-106b~25 or knockout mice at either time point, although, interestingly, mice that received AAV9-miR-106b~25 had a slight but significant reduction in the percentage of binucleated cardiomyocytes at p12 ($61.2 \pm 2.0\%$; **Fig.4d**).

Finally, the total number of cardiomyocytes at p12 was considerably larger in mice that received AAV9-miR-106b~25 as were the number of EdU+ cardiomyocytes at p6 and p12 (**Fig.4e,f**). In addition, the number of EdU+ cardiomyocytes was lower at p12 in *miR-106b~25* knockout mice, indicating a requirement of the endogenous *miR-106b~25* cluster for cardiomyocyte proliferation in juvenile hearts (**Fig.4f**).

Taken together, these new experiments answer all questions this Reviewer posed. First, the new data fully supports the contention that elevating the expression of the *miR-106b~25* cluster drives CM proliferation, resulting in a higher number of CMs. Secondly, *miR-106b~25* knockout mice had similar LV volume, CM density, CM nucleation and the total number of CMs as wild-type juvenile mice. We could, however, observe that *miR-106b~25* knockout mice displayed a small but significant reduction in the number of proliferating CMs. We conclude from these data that the endogenous *miR-106b~25* cluster is - at least partly - required for the endogenous CM cell cycle in the postnatal phase.

Q6 *"The images provided for CM proliferation in myocardium appear somewhat limited as they leave some room for interpretation as to the cell type the respective nucleus is attributed to. e.g. what do the three arrows in 3H point to compared to all the other red nuclei? There is a considerable number of yellow nuclei, suggesting an overlay of green (alpha actinin) and red (EdU) - aren't those the proliferating CMs?"*

Q7 *"Also Fig. 6J appears limited in this respect"*

Q8 *"Both Fig. 3 and Fig. 6 deserve higher magnification pictures (and possibly thinner confocal imaging sections) that allow for better attribution of the signals to cardiomyocytes and other cell types"*

Authors' response Q6, Q7, Q8: We thank the Reviewer for the valuable suggestions. We performed new confocal microscopy with thinner sections and acquired images with higher magnification so that nuclear identities are better interpretable. Examples of improved Figures are inserted below from **Fig.3h** and **Fig.6h** in the revised manuscript, but similar higher quality images were also generated for **Suppl.Fig.5d,f**. The arrows in **Fig.3h** simply highlight examples of EdU+ CMs and were not meant to indicate exceptions.

Figure 3

Figure 6

Q9 "Fig S1 E and F deserve better labelling and detailed description in this respect, e.g. upper panels are control, lower are AAV-miR, colour coding and numbering of P1, P2 and P3 in confusing"

Q10 "[Fig S1 F]... How many mice were studied?"

Q11 "What is the actual percentage of adult cardiomyocytes positive for Edu? Does the 'significantly increased' refer to the 0.2%, is this the average? How does this compare to the results of the confocal analyses?"

Authors' response Q9, Q10, Q11: Your points are well taken.

(Q9) We have now labeled the mentioned Figure (now new **Suppl.Fig.4a**). We also defined P1, P2 and P3 more clearly.

(Q10) Number of mice in new **Suppl.Fig.4b** belongs to the *in vivo* study performed and quantified in **Fig.3e-j** and amounts to n=3 animals per group.

(Q11) The percentages EdU+ CMs from the FACS analysis is 0,6 and 1.3%, which represents a doubling. From our confocal analysis the percentages are $4.6 \pm 0.4\%$ for hearts receiving AAV9-MCS and $7.5 \pm 0.7\%$ for hearts receiving AAV9-miR-106b~25 (**Fig.3i**). The techniques to assess proliferating cells are, of course, quite different; whereas FACS depends on a subset of cells following dissociation of an intact heart, counting from confocal microscopy analyzes sections of an complete heart. We value the inclusion of different techniques to assess the same phenomenon, and we are reassured by the conclusion as both independent techniques demonstrate a doubling in percentage of proliferating cardiomyocytes. Moreover, we think the studies described in **Fig.4** now independently support the conclusion that AAV9-miR-106b~25 stimulates CM proliferation.

Q12 "The authors should discuss the time course of the downregulation of the miR cluster in relation to the reported loss of proliferation capacity. It appears that 93 and 25 are lost relatively late in this respect and 106b may then be the responsible one? Is the downregulation occurring in CMs as opposed to other cell types?"

Authors' response Q12: Your point is well taken. We have performed a Langendorff free method to dissect cell types from the adult mouse heart (**Suppl.Fig.3**) to quantify the expression level of each cluster member in primary adult cardiomyocytes (CM), endothelial cells (EC) or fibroblasts (FB) (see image pasted below).

The data show – as we expected – that the expression of the endogenous *miR-106b~25* cluster in adult heart muscle cells is very low as compared to the non-myocyte cell fraction that still have a capacity to proliferate. On average, we have no experimental indication that one specific cluster member is more potent than the others in evoking proliferation either *in vitro* (**Fig.3c,d**) or *in vivo* (**Suppl.Fig.5**). Direct overexpression of either *miR-106b*, *miR-93* or *miR-25* *in vitro* by mimics transfection of each miRNA (**Fig.3c,d**), or *in vivo* by individually created AAV9 vectors (**Suppl.Fig.5**) does not show discernable differences in proliferation between the 3 cluster members.

Q13 "The Hippo pathway findings should be discussed with respect to the recent literature (Nature paper by J. Martin group etc.) as should the findings by the Mercola group on miR-25"

Authors' response Q13: We have followed the Reviewer's suggestion and included these phrases to the **Discussion section** (page 11,12) in relationship to the Hippo pathway and the findings by the Mercola group on miR-25.

"... That members of the *miR-106b~25* cluster can evoke cardiomyocyte proliferation is confirmed by an unbiased, high-content screen to identify proliferative microRNAs,³⁶ while more recently, *miR-25* was demonstrated to provoke cardiomyocyte proliferation in zebrafish by repressing the cell cycle inhibitor *Cdknc1* and tumor suppressor *Lats2*.³⁷ Our results also revealed components of the Hippo/Yap pathway as *miR-106b~25* targetome members. Hippo signaling has been widely studied in context of cardiac regeneration.^{38,39} In line, embryonic overexpression of *Yap* in mice induces hyperproliferation of cardiomyocytes and severely disproportional ventricles and death,^{38,40,41} while forced expression of *Yap* in the adult heart provokes cardiomyocyte cell cycle re-entry and regeneration postinfarction injury.^{40,42} However, unrestrained *Yap* activation may also display unwanted effects in pressure overloaded hearts due to cardiomyocyte dedifferentiation.⁴³

Interestingly, contradicting effects of miR-25 in the rodent heart have been reported. Some reports indicate that inhibition of miR-25 expression can lead to derepression of the target gene *Serca2a* and improve cardiac function,⁴⁴ and others report protection against oxidative stress or apoptosis induced by sepsis.^{45,46} In contrast, others report that overexpression of miR-25 is innocuous and induces proliferation by altering cell cycle genes in zebrafish,³⁷ while here we report cardiac enlargement secondary to enhanced cardiomyocyte proliferation, which at first sight could be misinterpreted as a pathological phenotype. From a therapeutic perspective, miR-25 loss-of-function approaches have also shown disparate results from improving contractility on the one side,⁴⁴ or inducing high blood pressure,⁴⁷ atrial fibrillation,⁴⁸ eccentric remodeling and dysfunction,^{20,47} on the other. It should be noted that distinct chemistries of antisense oligonucleotides can show quite different specificity or even cause side-effects that may explain the opposing observations.^{49,50} To avoid the uncertainty surrounding the use of oligonucleotide chemistries, here we resorted to an unequivocal gene deletion strategy where miR-106b~25 null mice display pathological cardiomyocyte hypertrophy, fibrosis, cardiac dilation and dysfunction, phenotypes that were recapitulated when silencing the individual cluster members with a 2'OME antisense chemistry. Using the same gene deletion approach, miR-106b~25 knockout mice show enhanced paroxysmal atrial fibrillation related to disruption of a paired-like homeodomain transcription factor 2 homeobox gene (*Pitx2*) driven mechanism that controls the expression of the miR-17~92 and miR-106b~25 clusters.⁵¹ In line, *Pitx2* lies in close proximity to a major atrial fibrillation susceptibility locus on human chromosome 4q25 identified in genome-wide association studies.⁵² Taken together, exceptional scrutiny should be considered when designing silencing strategies to therapeutically intervene in miR-25 expression in heart disease...."

Minor

Q14 "Scale bars are sometimes missing"

Q15 "Mice are not 'treated with AAV for 4 weeks' but studied 4 weeks after AAV injection"

Authors' response Q14, Q15: We have followed the Reviewer's suggestion and repaired the missing scale bars in several Figures where it applies to and corrected the sentence to "studied 4 weeks after AAV injection" throughout the manuscript (marked in **yellow**).

Reply to Reviewer #2 (Revision 1)

Dirkx, Raso et al. NCOMMS-17-20364. "A microRNA program controls the transition of cardiomyocyte hyperplasia to hypertrophy and stimulates mammalian cardiac regeneration".

We thank Reviewer #1 for his/her helpful and constructive remarks that improved the manuscript. We have carefully considered all of the suggestions proposed and revised the manuscript accordingly. Please note that all changes in the manuscript are marked in yellow.

Below the Reviewer will find a point-by-point rebuttal to his/her questions (Q).

Q1 "The model in Figure 1 appears to have fundamental problems. Before the stress is given, the mouse heart already has a significant phenotype. It is unclear whether the phenotype after TAC is due to the function of the miR-106b~25 cluster during development or during pressure overload stress. For example, if the heart has less myocytes before TAC, it alone may exacerbate cardiac dysfunction due to reduced contractility"

Q2 "The cardiac phenotype is poorly investigated in Figure 1. For example, the authors could have investigated the total number of cardiomyocytes in the heart and the extent of cell death".

Authors' response Q1, Q2: Your points are well taken. **Q1:** It is indeed correct that the sham miR-106b~25 knockout heart already showed a phenotype and this was emphasized in the text. However, we felt we could only answer all questions by employing stereology to simultaneously assess left ventricular volumes, cardiomyocyte (CM) volumes, CM nuclei densities, CM nucleation and CM proliferation as described previously (Alkass et al. Cell. 2015; Sampaio-Pinto V et al. Stem Cell Reports. 2018). In addition, we decided to analyze mouse hearts in the early postnatal phase to address the reviewer's valid question whether the endogenous miR-106b~25 cluster controls CM cell cycle, since CM proliferation is still abundant in early postnatal life in the mouse and is essentially absent after postnatal day 15 (see e.g. Alkass et al. Cell. 2015).

Accordingly, we included 3 experimental groups: (a) wild-type neonatal mice, (b) wild-type neonatal mice injected AAV9-miR-106b~25 intraperitoneally at p1 to elevate cardiac miR-106b~25 cluster expression, and (c) miR-106b~25 cluster knockout mice. We administered EdU intraperitoneally at p1, p2 and p9 in all groups to mark proliferating cells and analyzed hearts at p6 and p12 (**Fig.4a**).

LV volumes, cardiomyocyte volumes and total number of cardiomyocytes were determined as described previously (Alkass et al. Cell. 2015; Sampaio-Pinto V et al. Stem Cell Reports. 2018). Cardiomyocyte nuclei in tissue sections were unequivocally identified by staining for the cardiomyocyte nuclear marker pericentriolar material 1 (PCM-1). Cardiomyocyte multinucleation was determined by co-staining with wheat germ agglutinin (WGA) to delineate cell boundaries and PCM-1 for cardiomyocyte nuclei. Cardiomyocyte proliferation was assessed by co-labelling cardiac sections with PCM-1 and EdU (**Fig.4b**).

In neonatal mice, LV volumes increased from $5.5 \pm 0.3 \text{ mm}^3$ to $13.2 \pm 0.8 \text{ mm}^3$, with no substantial differences when mice received AAV9-miR-106b~25 or mice that were genetically deficient for the cluster, although there was a tendency for mice that received AAV9-miR-106b~25 to have a slightly higher LV volume (**Fig.4c**).

The ratio of mono- to binucleated cardiomyocytes changed substantially during the first 12 postnatal days. On p6, a minority of cardiomyocytes ($14.1 \pm 1.1\%$) were binucleated, whereas at p12 the majority of cardiomyocytes became binucleated ($66.1 \pm 0.9\%$), with no substantial differences with mice that received AAV9-miR-106b~25 or knockout mice at

either time point, although, interestingly, mice that received AAV9-miR-106b~25 had a slight but significant reduction in the percentage of binucleated cardiomyocytes at p12 ($61.2 \pm 2.0\%$; **Fig.4d**).

Finally, the total number of cardiomyocytes at p12 was considerably larger in mice that received AAV9-miR-106b~25 as were the number of EdU+ cardiomyocytes at p6 and p12 (**Fig.4e,f**). In addition, the number of EdU+ cardiomyocytes was lower at p12 in *miR-106b~25* knockout mice, indicating a requirement of the endogenous *miR-106b~25* cluster for cardiomyocyte proliferation in juvenile hearts (**Fig.4f**).

Taken together, these new experiments answer all questions this Reviewer posed. First, the new data fully supports the contention that elevating the expression of the *miR-106b~25* cluster drives CM proliferation, resulting in a higher number of CMs. Secondly, *miR-106b~25* knockout mice had similar LV volume, CM density, CM nucleation and the total number of CMs as wild-type juvenile mice. We could, however, observe that *miR-106b~25* knockout mice displayed a small but significant reduction in the number of proliferating CMs. We conclude from these data that the endogenous *miR-106b~25* cluster is - at least partly - required for the endogenous CM cell cycle in the postnatal phase.

Q3 "Does upregulation of *Mef2D* play a significant role in mediating the cardiac phenotype? Is the cardiac phenotype in Figure 1 caused by pathological hypertrophy, the lack of cardiomyocytes proliferation, or other mechanisms?"

Authors' response Q3: This is an insightful question. The Reviewer asks whether in the overexpression studies of the *miR-106b~25* cluster, repression of the pro-hypertrophic validated targets *Hand2* and *Mef2d* also participate in re-activation of the cardiomyocyte cell cycle. First, loss-of-function studies for *Mef2d* (Kim et al. J Clin Invest. 2008) or *Hand2* (which we performed in earlier, see e.g. Dirx et al. Nat Cell Biol. 2013) indicate no evidence

for a potential involvement of either factor in proliferation of postnatal cardiomyocytes, but rather confirm their function in cardiac hypertrophy *in vivo*. Secondly, we performed extensive literature searches to obtain any indication whether or not GO terms of “cell cycle activity” or “proliferation” have been assigned to Mef2d and Hand2, but neither have such indication. Finally, from our stereology experiments (**Fig.4**) where we simultaneously measured left ventricular volumes, cardiomyocyte (CM) volumes, CM nuclei densities, CM nucleation and CM proliferation in hearts from *miR-106b~25* KO mice where Mef2d is reactivated, we observed that LV volume, CM density, CM nucleation and the total number of CMs was equal to wild-type mice. The combined evidence points to a role for the *miR-106b~25* target gene Mef2d in cardiomyocyte hypertrophy.

As the Reviewer can perceive from **Fig.5a,e,f**, we have now performed an unbiased high-content screen in neonatal rat cardiomyocytes with siRNAs against 18 targets of the *miR-106b~25* cluster to screen targetome members for their individual contribution to cardiomyocyte proliferation. We have dedicated our resources (3 siRNAs designed against each target for a total of 18 targets in the unbiased screen is a remarkably expensive undertaking) only to *miR-106b~25* cluster targets for which stronger evidence of involvement in cell cycle re-activation could be expected. The data demonstrate that treatment with siRNAs against targetome members induced only a partial increase in CM proliferation compared to that observed with *miR-106b~25* overexpression, indicating that the effect of the miRNA cluster probably results from a cumulative effect on multiple, cellular mRNA targets.

Q4 *"Similarly, no mechanistic experiment has been conducted to elucidate the mechanism by which the heart treated with AAV9 miR-106b, AAV9 miR-93, or AAV9 miR-6 is enlarged. What is the total cell number in the heart? Similarly, what is the effect of AAV9 miR-106b~25 upon the total cell number in the heart?"*

Authors' response Q4: We refer to our answers to **Q1, Q2** above.

Q5 *"Although the authors have suggested potential targets in Figure 5, no evidence was provided regarding their mechanistic involvement (significance)"*

Authors' response Q5: We refer to our answers to **Q3** above.

Q6 *"Again, the analysis of cardiac phenotype is poorly conducted in Figure 6. What is the cell size or the total cell number in the heart? The data shown in Figure 6j is insufficient as evidence of cardiomyocyte proliferation. Can we really say that the rescue of cardiac function by AAV9-miR106b~25 is mediated through cardiomyocyte proliferation?"*

Authors' response Q6: To answer the Reviewer's question, we performed the following experiments to better phenotype the animal models.

First, as answered to this Reviewer's **Q1, Q2** above, we felt we could only answer these questions by employing stereology to simultaneously measure left ventricular volumes, cardiomyocyte (CM) volumes, CM nuclei densities, CM nucleation and CM proliferation. As explained above, the new data supports the contention that elevating the expression of the *miR-106b~25* cluster drives the CM cell cycle, resulting in a higher number of proliferating CMs and a higher total number of CMs (**Fig.4**).

In a second major approach, we cross-bred tamoxifen-inducible genetic lineage-tracing Myh6-MerCreMer mice to a Rosa26 tdTomato reporter model to permanently mark Myh6-expressing cells (i.e. cardiomyocytes) and follow the fate of their cellular descendants *in vivo*. Accordingly, adult Myh6-MCM x R26tdTomato animals were treated with tamoxifen daily for 7 days to label cardiomyocytes with tdTomato. Next, Myh6-MCM x R26tdTomato mice underwent permanent ligation of left anterior descending (LAD) coronary artery to induce myocardial infarction (MI) and hearts were injected in the peri-infarcted area with AAV9-miR-106~25 or a control AAV9 vector. A week before sacrifice, all animals received Edu to mark nuclei that are in S phase of the cell cycle and analyzed Edu+/tdTomato+ cells in tissue sections. As expected, tdTomato was exclusively expressed in cardiomyocytes. Importantly, the number of Edu+/tdTomato+ cells in post-infarcted hearts that received AAV9-miR-106~25 was doubled compared to post-infarcted hearts that received a control AAV9 vector. This genetic lineage tracing approach confirms that the AAV9-miR-106~25 vector stimulated proliferation of pre-existing cardiomyocytes (**Fig 5i,j,k**).

Q7 "A paper published in *Nature* (508, 531-535, 2014) reports an opposite function of miR-25 in the heart. The authors could have discussed this issue"

Authors' response Q7: We agree with the Reviewer's suggestion and included these phrases to the Discussion section (page 11,12) in relationship to the findings by the Mercola group on miR-25.

"... Interestingly, contradicting effects of miR-25 in the rodent heart have been reported. Some reports indicate that inhibition of miR-25 expression can lead to derepression of the target gene *Serca2a* and improve cardiac function,⁴⁴ and others report protection against oxidative stress or apoptosis induced by sepsis.^{45,46} In contrast, others report that overexpression of miR-25 is innocuous and induces proliferation by altering cell cycle genes in zebrafish,³⁷ while here we report cardiac enlargement secondary to enhanced cardiomyocyte proliferation, which at first sight could be misinterpreted as a pathological phenotype. From a therapeutic perspective, miR-25 loss-of-function approaches have also shown disparate results from improving contractility on the one side,⁴⁴ or inducing high blood pressure,⁴⁷ atrial fibrillation,⁴⁸ eccentric remodeling and dysfunction,^{20,47} on the other. It should be noted that distinct chemistries of antisense oligonucleotides can show quite different specificity or even cause side-effects that may explain the opposing observations.^{49,50} To avoid the uncertainty surrounding the use of oligonucleotide chemistries, here we resorted to an unequivocal gene deletion strategy where miR-106b~25 null mice display pathological cardiomyocyte hypertrophy, fibrosis, cardiac dilation and dysfunction, phenotypes that were recapitulated when silencing the individual cluster members with a 2' Ome antisense chemistry. Using the same gene deletion approach, miR-106b~25 knockout mice show enhanced paroxysmal atrial fibrillation related to disruption of a paired-like homeodomain transcription factor 2 homeobox gene (*Pitx2*) driven mechanism that controls the expression of the miR-17~92 and miR-106b~25 clusters.⁵¹ In line, *Pitx2* lies in close proximity to a major atrial fibrillation susceptibility locus on human chromosome 4q25 identified in genome-wide association studies.⁵² Taken together, exceptional scrutiny should be considered when designing silencing strategies to therapeutically intervene in miR-25 expression in heart disease."

Reply to Reviewer #3 (Revision 1)

Dirkx, Raso et al. NCOMMS-17-20364. "A microRNA program controls the transition of cardiomyocyte hyperplasia to hypertrophy and stimulates mammalian cardiac regeneration".

We thank Reviewer #1 for his/her helpful and constructive remarks that improved the manuscript. We have carefully considered all of the suggestions proposed and revised the manuscript accordingly. Please note that all changes in the manuscript are marked in yellow.

Below the Reviewer will find a point-by-point rebuttal to his/her questions (Q).

Q1 "The authors claimed that end-stage HF animal models were used in entire in-vivo experiment. However, there is no molecular evidence to support this claim. Rather, based on the functional characteristics the authors provided in this manuscript seems more close to hypertrophic state. Therefore, it is highly recommended to provide the miRNA profiling including miRNA-106b~25 cluster during cardiac hypertrophy (eg. 1 or 2wks post TAC will be good model). Since expression pattern of many signature molecules are substantially differed in hypertrophy and adverse heart failure these additional studies are important"

Authors' response: We thank this Reviewer for the suggestion. We agree it is indeed important to discern (compensated) hypertrophic phases of the heart versus end-stage heart failure with "clinical" and molecular evidence of failure in the models.

Accordingly, we performed a complete new study where we subjected mice to transverse aortic constriction (TAC) or sham surgery for a period of either 1, 2 and 6 weeks, performed echocardiography and analyzed expression of molecular markers of heart failure and the expression of each cluster member separately.

Supplemental Figure 1c

Supplemental Figure 1b

As the Reviewer correctly suggested, hearts subjected to pressure overload for 1 or 2 weeks show a hypertrophic response, show no dilation (increase in LVIDs), have sustained contractile function (no decrease in %EF), did show a mild increase in the sensitive fetal gene

marker beta-myosin heavy chain (*Myh7*) and a significant increase in the clinical biomarker BNP (*Nppb*). Remarkably, even in this early phase all members of the *miR-106b~25* cluster showed a pronounced decrease in expression. Furthermore, murine hearts subjected to 6 weeks of sustained pressure overload displayed dilation, loss of contractile function, significant increases in the expression of markers genes *Myh7* and *Nppb* and a similarly pronounced decrease in expression of all cluster members. These data are now incorporated in **Suppl.Fig.1b,c**.

Supplemental Figure 1

Figure 1

In the original manuscript, we already provided evidence of reduced expression of the members of the *miR-106b~25* cluster in hearts/biopsies of end-stage heart failure in two “models”: **1)** transgenic mice with constitutive overexpression of an activated mutant of

calcineurin that typically display extensive fibrosis, an EF of less than 20%, ventricular dilation and very high expression of all molecular markers of heart failure (e.g. De Windt LJ et al... Molkenkin JD. Circ Res 2000) (**Suppl.Fig.1a**); and **(2)** in biopsies of explanted hearts from end-stage heart failure patients who required a heart transplantation (**Fig.1b**).

Taken together, these new experiments answer all questions this Reviewer posed. The data show that both in early hypertrophic phases of cardiac remodeling as well in biopsies of murine and human hearts in end-stage heart failure, the *miR-106b~25* cluster is reduced in expression.

Q2 "A major inconsistency in the study is that Dr. Mauro Giacca's group showed in their paper (Eulalio et al, Nature 2012) that miR-25 is one of the miRNAs that DECREASE cardiomyocyte proliferation (Supplementary figure 2 in previous Nature paper (Eulalio et al., 2012)). The data of Dr Giacca (an author on this study) directly contradict the authors work"

Authors' response: This question is based on a common misconception of the stem loop structure of microRNAs. To clarify this, we visualized this structure for the Reviewer below.

Here, a typical precursor microRNA is displayed that will give rise to two mature microRNA sequences each ~ 22 nt in length following Drosha and Dicer processing, one commonly referred to a major or -5p strand, the second referred to as minor or -3p or star strand (**panel a** left). The mature miR-25 this manuscript deals with is miR-25-5p or the major strand (**panel b** left);

SUPPLEMENTARY INFORMATION doi:10.1038/nature13179

microRNA	miRBase mature sequence accession	Sequence	RAT			MOUSE
			total number cells analyzed	% cardiomyocytes (α-actinin+)	% proliferating cardiomyocytes (α-act+, Ki-67+, EdU+)	% proliferating cardiomyocytes (α-act+, Ki-67+, EdU+)
hsa-let-7i	MIMAT0000415	UGAGGUAGUAGUUUGUCUGUU	2856	86.97	0.95	-
hsa-miR-25	MIMAT0000081	CAUJUGCACUJUGUCGGUCUGA	3410	80.54	12.75	-
hsa-miR-25*	MIMAT0004498	AGCGGAGACUJGGCAAUUG	2206	84.54	0.16	-
hsa-miR-93	MIMAT0000093	CAAAGUCGUGUCGUCAGGUAG	3261	84.91	28.60	6.21
hsa-miR-106b	MIMAT0000680	UAAAGUCGUCAGAGUCAGAU	3201	87.39	27.52	7.16

Fig.1a manuscript). It's mature nucleotide sequence is also written in full. Finally, given this misconception, the miR-25 star strand that the Reviewer refers to from the paper by Eulalio et al. Nature 2012 indeed doesn't evoke proliferation, but the major strand, the topic of this study, actually does (12.75% proliferating cardiomyocytes compared to e.g. hsa-let-7i with 0,95%, essentially no proliferative capacity, **panel c** below). In the same supplementary dataset by Eulalio et al., we also found independent confirmation that the other members of the miR-106b~25 cluster, miR-93 and miR-106b, evoke proliferation of cardiomyocytes (indicated by **red arrows**).

Conclusively, from the high-content screen performed by Eulalio et al., we obtain independent confirmation that all members of the *miR-106b~25* cluster, including miR-25-5p, provoke cardiomyocyte proliferation when overexpressed.

Q3 "All histological data (MT and Picrosirius red etc) show that the size of cardiomyocytes from AAV9-miR-106b~25 injected mouse hearts seem much larger than those of control group. It is unclear how authors measure CSA in this paper (Fig 2d and 3f)"

Authors' response Q3: Your point is well taken. We do not use Sirius red stained images to quantify cardiomyocyte cell surface area as the boundaries nor the nuclei of cardiomyocytes are visible with this type of staining and only use Sirius red stained to visualize and quantify interstitial or perivascular fibrosis. To quantify cross-sectional cell surface areas of cardiomyocytes, mouse hearts were arrested in diastole, perfusion fixed with 4% paraformaldehyde/PBS solution, embedded in paraffin, sectioned at 4 μm and stained with FITC-conjugated Wheat Germ Agglutinin (WGA) to visualize cell membranes and quantify the myocyte cross-sectional area using ImageJ imaging software as described several times by us and others (refs 20, 53, 57, 58 and e.g. Bensley et al. *Sci Rep.* 2016;6:23756). To aid in the visualization of differences in cell surface areas, we pasted below the images of WGA-stained sections reported in **Fig.2d**, **Fig.3f** and **Fig.1d** of the revision, showing no discernable differences between the experimental groups in **Fig.2d** and **Fig.3f**. In contrast, **Fig.1d** shows clearly hypertrophic cardiomyocytes in mice subjected to transverse aortic constriction (TAC).

Fig.2d

Fig.3f

Fig.1d

Q4 "... In Fig.1, the authors show that knockout of the *miR-106b~25* cluster leads to mild cardiac hypertrophy and fibrosis at baseline. This data is basically same as the Figure 6 of the author's previous paper (*Nature Cell Biology*, 2013). In this previous study, they showed that knock-down of *miR-25* using an antagomir led to similar cardiac dysfunction at baseline. Thus, the current Fig 1 does not tell anything about the functional significance of *miR-106b* and *miR-93*. It would have been much more informative on the role of *miR-106b* and *miR-93*, if they performed similar knockdown experiments with individual antagomirs of *miR-106b* or *miR-93*. In Fig 2, they observed the gain-of-function effects of the individual *miRs*. Then, it is not clear why they did not perform similar experiments for the loss-of-function effects of the individual *miRs*?"

Authors' response Q4: At the request of this Reviewer, we have now performed a new and very elaborate study where we designed so-called "antagomirs" (fully complementary, antisense oligonucleotides 2'-OMe-modified RNA nucleotides with phosphorothioate linkages and a cholesterol moiety linked through a hydroxyprolinol linkage at the 5' end) to silence either *miR-25*, *miR-93* or *miR-106b*. To obtain robust enough numbers, we divided the animals to receive either a control antagomir (n=10), or an antagomir designed to specifically silence *miR-25* (n=10, essentially the same as we described previously in ref. 20, Dirx et al. *Nat Cell Biol* 2013), *miR-93* (n=10) or *miR-106b* (n=10).

The data are predictable given the cardiac phenotype of the *miR-106b~25* knockout mouse described in this manuscript. Indeed, single silencing of either *miR-25*, *miR-93* or *miR-106b* produced spontaneous hypertrophic growth with histological abnormalities, accompanied with induction of fetal gene expression, and cardiac dysfunction as measured by echocardiography, but always to a lesser extent than observed in the *miR-106b~25* knockout mice. These data fully support the contention that the *miR-106b~25* cluster is co-transcribed and where each cluster

member functions synergistically with the other members to produce the same phenotype when reduced in expression (sustain hypertrophic growth). The new data are presented in **Suppl.Fig.2**.

Q5 *"In page 5, line 4, they wrote "overexpression of the miR clusters to physiological levels". What do they mean by "physiological level"? Do they mean the level at p0? It is also unclear whether the expression level was indeed restored to the levels seen at p0 or it was much more up-regulated beyond the p0 level. For clarification, Fig 2C is to be revised to include the expression level of individual miR at p1 when the viruses were injected. It is extremely important to discern whether the proliferative effects of the miR are seen when the expression level was merely restored to the level seen at p0 or when it is elevated to a non-physiologically high level"*

Authors' response Q5: Your point is well taken. We corrected the sentence to: *"...Conclusively, maintaining high miR-106b~25 expression levels as observed in the early postnatal developmental period produced cardiac enlargement in the adult heart without classical signs of pathological hypertrophic remodeling..."* (page 5, Results).

We also felt we could only unequivocally answer this question by employing stereology to simultaneously assess left ventricular volumes, cardiomyocyte (CM) volumes, CM nuclei densities, CM nucleation and CM proliferation as described previously (Alkass et al. Cell. 2015; Sampaio-Pinto V et al. Stem Cell Reports.2018). In addition, we decided to analyze mouse hearts in the early postnatal phase to address the reviewer's valid question whether the endogenous miR-106b~25 cluster controls CM cell cycle, since CM proliferation is still abundant in early postnatal life in the mouse and is essentially absent after postnatal day 15 (see e.g. Alkass et al. Cell. 2015).

Accordingly, we included 3 experimental groups: (a) wild-type neonatal mice, (b) wild-type neonatal mice injected AAV9-miR-106b~25 intraperitoneally at p1 to elevate cardiac miR-106b~25 cluster expression, and (c) miR-106b~25 cluster knockout mice. We administered EdU intraperitoneally at p1, p2 and p9 in all groups to mark proliferating cells and analyzed hearts at p6 and p12 (**Fig.4a**).

LV volumes, cardiomyocyte volumes and total number of cardiomyocytes were determined as described previously (Alkass et al. Cell. 2015; Sampaio-Pinto V et al. Stem Cell Reports. 2018). Cardiomyocyte nuclei in tissue sections were unequivocally by staining for the cardiomyocyte nuclear marker pericentriolar material 1 (PCM-1). Cardiomyocyte multinucleation was determined by co-staining with wheat germ agglutinin (WGA) to delineate cell boundaries and PCM-1 for cardiomyocyte nuclei. Cardiomyocyte proliferation was assessed by co-labelling cardiac sections with PCM-1 and EdU (**Fig.4b**).

In neonatal mice, LV volumes increased from 5.5 ± 0.3 mm³ to 13.2 ± 0.8 mm³, with no substantial differences when mice received AAV9-miR-106b~25 or mice that were genetically deficient for the cluster, although there was a tendency for mice that received AAV9-miR-106b~25 to have a slightly higher LV volume (**Fig.4c**). The ratio of mono- to binucleated cardiomyocytes changed substantially during the first 12 postnatal days. On p6, a minority of cardiomyocytes ($14.1 \pm 1.1\%$) were binucleated, whereas at p12 the majority of cardiomyocytes became binucleated ($66.1 \pm 0.9\%$), with no substantial differences with mice that received AAV9-miR-106b~25 or knockout mice at either time point, although mice that received AAV9-miR-106b~25 had a slight but significant reduction in the percentage of binucleated cardiomyocytes at p12 ($61.2 \pm 2.0\%$; **Fig.4d**).

Finally, the total number of cardiomyocytes at p12 was considerably larger in mice that received AAV9-miR-106b~25 as were the number of EdU+ cardiomyocytes at p6 and p12 (**Fig. 4e,f**). In addition, the number of EdU+ cardiomyocytes was lower at p12 in *miR-106b~25* knockout mice, indicating a requirement of the endogenous *miR-106b~25* cluster for cardiomyocyte proliferation in juvenile hearts (**Fig. 4f**).

Taken together, these new experiments answer all questions this Reviewer posed. First, the new data fully supports the contention that elevating the expression of the *miR-106b~25* cluster drives CM proliferation, resulting in a higher number of CMs. Secondly, *miR-106b~25* knockout mice had similar LV volume, CM density, CM nucleation and the total number of CMs as wild-type juvenile mice. We could, however, observe that *miR-106b~25* knockout mice displayed a small but significant reduction in the number of proliferating CMs. We conclude from these data that the endogenous *miR-106b~25* cluster is - at least partly - required for the endogenous CM cell cycle in the postnatal phase.

Q6 "In Fig 3a~d, the authors utilized cardiomyocytes isolated from neonatal hearts. In neonatal hearts, the expression level of each miR is high. Why would they express more miRs on top of the already high level of miR? Once again, this reviewer suspect that the proliferative effects of miRs are achieved when the miRs are overexpressed beyond the physiological level. To address this issue, the authors should use cardiomyocytes isolated from adult hearts where the expression levels of miR hit bottom"

Authors' response Q6: As answered to this Reviewer's Q5 above, we corrected the sentence to: "...Conclusively, maintaining high *miR-106b~25* expression levels as observed in the early postnatal developmental period produced cardiac enlargement in the adult heart without classical signs of pathological hypertrophic remodeling..." (page 5, Results). The point of that experiment was to maintain higher *miR-106b~25* expression towards adulthood comparable to the neonatal period when cardiomyocytes still retain substantial proliferative capacity.

Secondly, we already performed the experiment in heart muscle cells when *miR-106b~25* expression “hits bottom” in **Fig.6a-h** (now also confirmed in **Suppl.Fig.3**). Indeed, **Fig.6a-h** describes the effects of intracardiac injection of AAV-miR-106b~25 in adult hearts of wild-type hearts.

Finally, in a major additional approach, we cross-bred tamoxifen-inducible genetic lineage-tracing Myh6-MerCreMer mice to a Rosa26 tdTomato reporter model to permanently mark Myh6-expressing cells (i.e. cardiomyocytes) and follow the fate of their cellular descendants *in vivo*. Accordingly, **adult** Myh6-MCM x R26tdTomato animals were treated with tamoxifen daily for 7 days to label cardiomyocytes with tdTomato. Next, Myh6-MCM x R26tdTomato mice underwent permanent ligation of left anterior descending (LAD) coronary artery to induce myocardial infarction (MI) and hearts were injected in the peri-infarcted area with AAV9-miR-106~25 or a control AAV9 vector. A week before sacrifice, all animals received Edu to mark nuclei that are in S phase of the cell cycle and analyzed EdU+/tdTomato+ cells in tissue sections. As expected, tdTomato was exclusively expressed in cardiomyocytes. Importantly, the number of EdU+/tdTomato+ cells in post-infarcted **adult** hearts that received AAV9-miR-106~25 was doubled compared to post-infarcted hearts that received a control AAV9 vector. This genetic lineage tracing approach confirms that the AAV9-miR-106~25 vector stimulated proliferation of **adult** pre-existing cardiomyocytes (**Fig 5i,j,k**).

Q7 "The same logics mentioned above (#3) applies to Fig 3e~j. The virus should be injected to adult mice and the expression level of miR should be carefully monitored. Once again, the important issue is whether the AAV-mediated expression of miR restores the lowered level of miRs to the level seen at p0 or to the level of way over the physiological level"

Authors' response Q7: We refer to our answer to **Q6** above.

Q8 ***"miR-25 is overexpressed in several cancer cells and miR-25 has been reported as a tumor suppressor via SOX4 and CyclinD1, or EZH2 activation especially on G1 phase of the cell cycle. (Chen et al., Tumour Biol., 2017; Xu et al., Asian Pac J Trop Med, 2013; Esposito et al., JCEM, 2012 etc). On the other hands, authors claim that miR-25 induces proliferating cardiomyocytes. This should be clarified"***

Authors' response Q8: The biomedical literature on *miR-25* is vast and often presents diametrically opposed findings on its expression and biological function in different cell types. Possibly the most recent and updated review on the topic can be found by Sarkozy et al. *A myriad of roles of miR-25 in health and disease*. *Oncotarget*. 2018;9:21580. The majority of these 72 studies to date mentioned in that review were performed in various cancers or cancer cell lines, with half of reports demonstrating increased *miR-25* expression and half of reports showing decreased *miR-25* expression in non-small cell lung cancer (NSCLC), breast-ovarian and prostate cancers, thyroid cancers and gastrointestinal cancers.

However, molecular mechanisms in cancer (cell lines) are vastly different from and can have opposing functions in heart muscle cells and should be extrapolated with great care. Thus, perhaps more instructive on the topic are reports on *miR-25* (and/or its co-expressed cluster members *miR-93* and *miR-106b*) in the heart, and more specifically heart muscle cells. Here, less than 10 publications are now available, yielding a limited body of literature that was discussed in the context of cardiomyocyte proliferation and hypertrophy in the new **Discussion section** (see **Q9** below).

Q9 ***"Other previous studies showed that miR25 is up-regulated in the failing hearts. The authors report the exact opposite observations. Any thoughts or explanations on this discrepancy? Better be mentioned in the discussion section"***

Authors' response Q9: We agree with the Reviewer's suggestion and included these phrases to the **Discussion section** (page 11,12) in relationship to the findings by the Mercola group on *miR-25*.

*"... Interestingly, contradicting effects of miR-25 in the rodent heart have been reported. Some reports indicate that inhibition of miR-25 expression can lead to derepression of the target gene *Serca2a* and improve cardiac function,⁴⁴ and others report protection against oxidative stress or apoptosis induced by sepsis.^{45,46} In contrast, others report that overexpression of miR-25 is innocuous and induces proliferation by altering cell cycle genes in zebrafish,³⁷ while here we report cardiac enlargement secondary to enhanced cardiomyocyte proliferation, which at first sight could be misinterpreted as a pathological phenotype. From a therapeutic perspective, miR-25 loss-of-function approaches have also shown disparate results from improving contractility on the one side,⁴⁴ or inducing high blood pressure,⁴⁷ atrial fibrillation,⁴⁸ eccentric remodeling and dysfunction,^{20,47} on the other. It should be noted that distinct chemistries of antisense oligonucleotides can show quite different specificity or even cause side-effects that may explain the opposing observations.^{49,50} To avoid the uncertainty surrounding the use of oligonucleotide chemistries, here we resorted to an unequivocal gene deletion strategy where *miR-106b~25* null mice display pathological cardiomyocyte hypertrophy, fibrosis, cardiac dilation and dysfunction, phenotypes that were recapitulated when silencing the individual cluster members with a 2' Ome antisense chemistry. Using the same gene deletion approach, *miR-106b~25* knockout mice show enhanced paroxysmal atrial fibrillation related to disruption of a paired-like homeodomain transcription factor 2 homeobox gene (*Pitx2*) driven mechanism that controls the expression of the *miR-17~92* and *miR-106b~25* clusters.⁵¹ In line, *Pitx2* lies in close proximity to a major atrial fibrillation susceptibility locus on human chromosome 4q25 identified in genome-wide association studies.⁵² Taken together, exceptional scrutiny should be considered when designing silencing strategies to therapeutically intervene in *miR-25* expression in heart disease."*

REVIEWERS' COMMENTS

Reviewer #1 (Remarks to the Author):

Review of revised manuscript by Raso et al.

Altogether, the authors have adequately addressed the major points that have been raised by this reviewer. In particular, several new experiments have been conducted, that deal with previous concerns and whose addition to the study has clearly strengthened the manuscript. The siRNA screen of putative miRNA targets and the quantification of total myocyte numbers (new Fig. 4) are particularly valuable data sets. The new data sets in the supplemental files are also appreciated, particular the FACS analysis and the experiments with the individual miRNAs. In fact, I may want to suggest that part of these convincing data (myocyte nuclei clearly visible and identifiable as belonging to myocytes) be added to the main figures, where the confocal pictures are still harder to assess for CM nucleus labelling.

With regard to the new data, the authors should provide a more information on the FACS experiment in the respective figure legend. The absolute numbers determined here for proliferating CMS appear quite different, as also discussed in the response to the reviewers. These arguments should also be included into the discussion section.

Reviewer #4 (Remarks to the Author):

the authors have adequately responded to the concerns of reviewer 3 with extensively more detailed experiments and clarifications in the text of the manuscript

Reply to Reviewer #1 (Revision 2)

Dirkx, Raso *et al.* NCOMMS-17-20364A-Z. "A microRNA program promotes the balance between cardiomyocyte hyperplasia to hypertrophy and stimulates cardiac regeneration".

We thank Reviewer #1 for his/her helpful and constructive remarks that improved the manuscript. We have carefully considered all of the suggestions proposed and revised the manuscript accordingly. Please note that all changes in the manuscript are marked in yellow.

Below the Reviewer will find a point-by-point rebuttal to his/her questions (Q).

Q1 *"...Altogether, the authors have adequately addressed the major points that have been raised by this reviewer. In particular, several new experiments have been conducted, that deal with previous concerns and whose addition to the study has clearly strengthened the manuscript. The siRNA screen of putative miRNA targets and the quantification of total myocyte numbers (new Fig. 4) are particularly valuable data sets. The new data sets in the supplemental files are also appreciated, particular the FACS analysis and the experiments with the individual miRNAs.*

In fact, I may want to suggest that part of these convincing data (myocyte nuclei clearly visible and identifiable as belonging to myocytes) be added to the main figures, where the confocal pictures are still harder to assess for CM nucleus labelling.

With regard to the new data, the authors should provide a more information on the FACS experiment in the respective figure legend. The absolute numbers determined here for proliferating CMS appear quite different, as also discussed in the response to the reviewers. These arguments should also be included into the discussion...."

Authors' response Q1: We are very grateful for the many compliments of this Reviewer on the revised manuscript.

We have followed the Reviewer's suggestion and now included the confocal images from EdU positive heart sections with individual miRNA overexpression studies from the Supplementary Information to **new main Figure 5h**.

In accordance with journal policy, we also now provided more information in the legend to the FACS experiment as well as an in-figure gating strategy to clarify the FACS experiment that now forms part of **Supplementary Figure 4** in the Supplementary Information.

Reply to Reviewer #4 (Revision 2)

Dirkx, Raso *et al.* NCOMMS-17-20364A-Z. "A microRNA program **promotes** the **balance between** cardiomyocyte hyperplasia to hypertrophy and stimulates cardiac regeneration".

We thank Reviewer #4 for his/her constructive remarks. Please note that all changes in the manuscript are marked in **yellow**.

Below the Reviewer will find a point-by-point rebuttal to his/her questions (Q).

Q1 *"...the authors have adequately responded to the concerns of reviewer 3 with extensively more detailed experiments and clarifications in the text of the manuscript..."*

Authors' response Q1: We are very grateful for the compliments of this Reviewer on the revised manuscript.